# Beyond accuracy: Tracking more like Human via Visual Search

**Dailing Zhang**[1,2]    **Shiyu Hu**[5]    **Xiaokun Feng**[1,2]
**Xuchen Li**[1,2]    **Meiqi Wu**[3]    **Jing Zhang**[2]    **Kaiqi Huang**[1,2,4]

[1]School of Artificial Intelligence, University of Chinese Academy of Sciences
[2]Institute of Automation, Chinese Academy of Sciences
[3]School of Computer Science and Technology, University of Chinese Academy of Sciences
[4]Center for Excellence in Brain Science and Intelligence Technology, Chinese Academy of Sciences
[5]School of Physical and Mathematical Sciences, Nanyang Technological University
zhangdailing2023@ia.ac.cn, shiyu.hu@ntu.edu.sg, fengxiaokun2022@ia.ac.cn,
wumeiqi18@mails.ucas.ac.cn, {lixuchen2024, jing_zhang, kqhuang}@ia.ac.cn

## Abstract

Human visual search ability enables efficient and accurate tracking of an arbitrary moving target, which is a significant research interest in cognitive neuroscience. The recently proposed Central-Peripheral Dichotomy (CPD) theory sheds light on how humans effectively process visual information and track moving targets in complex environments. However, existing visual object tracking algorithms still fall short of matching human performance in maintaining tracking over time, particularly in complex scenarios requiring robust visual search skills. These scenarios often involve **S**patio-**T**emporal **D**iscontinuities (*i.e.*, *STDChallenge*), prevalent in long-term tracking and global instance tracking. To address this issue, we conduct research from a human-like modeling perspective: (1) Inspired by the CPD, we propose a new tracker named **CPDTrack** to achieve human-like visual search ability. The central vision of CPDTrack leverages the spatio-temporal continuity of videos to introduce priors and enhance localization precision, while the peripheral vision improves global awareness and detects object movements. (2) To further evaluate and analyze *STDChallenge*, we create the ***STDChallenge Benchmark***. Besides, by incorporating human subjects, we establish a human baseline, creating a high-quality environment specifically designed to assess trackers' visual search abilities in videos across *STDChallenge*. (3) Our extensive experiments demonstrate that the proposed CPDTrack not only achieves state-of-the-art (SOTA) performance in this challenge but also narrows the behavioral differences with humans. Additionally, CPDTrack exhibits strong generalizability across various challenging benchmarks. In summary, our research underscores the importance of human-like modeling and offers strategic insights for advancing intelligent visual target tracking. Code and models are available at `https://github.com/ZhangDailing8/CPDTrack`.

## 1 Introduction

In the real world, humans excel at locating an arbitrary moving target within complex backgrounds and can resume tracking it even after temporary loss. This enduring question in cognitive neuroscience has recently been explained to some extent by the Central-Peripheral Dichotomy (CPD) theory, which suggests that the human eyes processes all visual inputs by sorting them into central and peripheral visions [1, 2]. Central vision, decoded and understood by higher brain areas, focuses on interpreting details and minimizing distractions; while peripheral vision, processed in the primary visual cortex (V1), swiftly detects dynamic changes.

38th Conference on Neural Information Processing Systems (NeurIPS 2024).

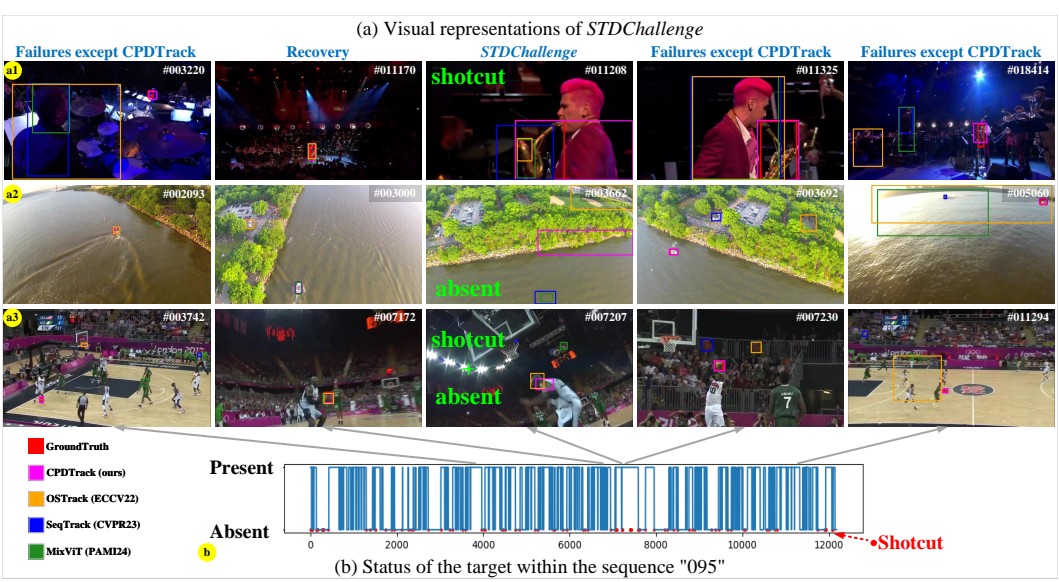

Figure 1: Illustration of the *STDChallenge*, depicting the *absent* of targets and *shotcut*. *STDChallenge* is quite challenging, but CPDTrack can maintain robust tracking performance, demonstrating stronger visual search abilities compared to other trackers. (a) In the first column, most trackers fail; in the second column, they can recover; in the third column, a *STDChallenge* occurs; in the fourth column, all trackers except for CPDTrack fail; and in the fifth column, this remains the case, meaning trackers' limited recovery. (b) shows the status of the target within the sequence of "095" from VideoCube [4], and red dots means *shotcut*.

Meanwhile, in computer vision research, advancements in single object tracking (SOT) are increasingly closing the gap between human dynamic visual ability (DVA) and proxy tasks [3, 4, 5]. In 2013, the short-term tracking (STT) task and related benchmarks were proposed [6, 7, 8]. However, the implicit assumption of continuous motion simplifies this task to continuously locating the target in short videos of tens of seconds, which is far from human DVA. Drawing on human visual search skills, the VOT community is broadening its scope by incorporating long-term tracking (LTT) [9, 10], which includes target *absent* (*i.e.*, the moving target can disappear and reappear in tracking), and global instance tracking (GIT) [4], which involves *shotcut* (*i.e.*, the video sequence may include different viewpoints and scenes). We refer to these challenges collectively as **spatio-temporal discontinuity (*STDChallenge*)**, which demands enhanced visual search abilities from trackers, as shown in Fig. 1.

Unfortunately, mainstream machines have not kept pace with the aforementioned expansion of task definitions to achieve human performance. Most mainstream trackers [11, 12, 13, 14, 15, 16], influenced by the definition of STT, depend on the spatio-temporal continuity of the target. These trackers typically perform local cropping based on the previous frame's results within the motion model, a module for calculating the search region on the current frame, as shown in Fig. 2(b-1). Obviously, this modeling mechanism only simulates human central vision, making it far from replicating human visual search ability, especially in scenarios where the *STDChallenge* increases significantly, as shown in Fig. 6(a). Several studies [17, 18] have also demonstrated that *STDChallenge* presents a significant challenge for trackers. Thus, a natural question is: **what causes trackers to overlook the importance of the *STDChallenge* in their modeling process**?

Some possible reasons from the benchmark perspective may answer the above question: **Limitation of datasets.** Although the definition of proxy tasks has been moving closer to human DVA (STT → LTT → GIT), most datasets are still influenced by the initial task characteristics (*i.e.*, *single-target*, *model-free*, *causal-trackers*, *single-camera*, and *short-term*) in data collection process [19, 20, 21, 9], as in Fig. 10, resulting in the importance of the *STDChallenge* being overlooked. Consequently, trackers can easily achieve good performance on most datasets without possessing sufficient visual search ability to handle the *STDChallenge*. **Limitation of evaluation.** Researchers contend that integrating insights from human DVA could address *STDChallenge* and enhance the real-world robustness of machines [22, 3, 23]. However, the substantial differences in methodologies and experimental

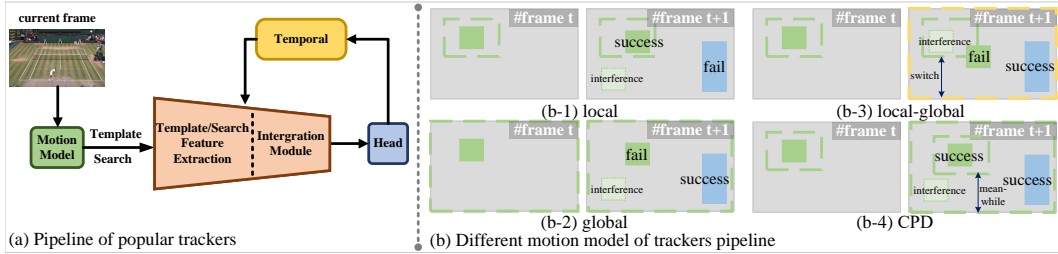

Figure 2: Comparison of tracking pipeline. Our CPDTrack differs from previous trackers in motion model. (a) The core components of mainstream tracking frameworks consist of a motion model, feature extraction, and a temporal module. (b-1) local tracker, which tracks targets in local areas and has difficulty recovering after failure; (b-2) global tracker, which tracks targets globally in the current frame, is susceptible to background interference and has lower efficiency; (b-3) local-global tracker, which can switch between the above two, depending on the performance of the local tracker; (b-4) our CPDTrack, which can model both central and peripheral information simultaneously.

approaches between computer science and cognitive science make it difficult to determine if current technological advancements are truly bridging the gap between SOT machines and human DVA. Consequently, there is a notable absence of mechanisms to assess the true visual intelligence of trackers.

In summary, existing trackers lack consideration of human visual search ability in their modeling mechanisms, making it difficult to cope with the *STDChallenge*. This limitation is not only due to the algorithm design itself but also involves the limitations of datasets and evaluation. Therefore, this work aims to address these aspects (algorithms and benchmark) and verifies the relevant results through experimental analysis.

**A new algorithm CPDTrack (Section 3).** Our analysis of the tracker pipeline and understanding of the CPD suggests that mainstream motion models, which crop local regions from the current frame, as shown in Fig. 2, restrict the tracker's ability to locate moving targets. Drawing on the distinction between central and peripheral vision in CPD, we propose a new tracker named CPDTrack to divide the current frame accordingly. Central vision captures detailed information, leveraging the video's spatio-temporal continuity to introduce priors, while peripheral vision manages global information to enhance overall scene understanding, as shown in Fig. 3. CPDTrack also models the information query, achieving top-down control based on cognitive outcomes.

**A new *STDChallenge Benchmark* with Visual Turing Test (Section 4).** To address the shortcomings of existing benchmarks environments, we introduce a specialized challenge environment named *STDChallenge Benchmark* to assess the visual search abilities of both machines and humans in tracking an arbitrary moving target. This environment comprises sequences that represent the *STDChallenge*, carefully selected from both LTT and GIT benchmarks. By sampling from a variety of benchmarks, we ensure a reduction in dataset-specific biases. Simultaneously, we employ the Visual Turing Test (VTT) [24, 25] on the *STDChallenge Benchmark* to assess the disparity in intelligence between trackers and human DVA. Specifically, we examine error consistency [26] to compare the performance of humans and machines, and explore behavioral differences among machines with varying architectures and parameters in dynamic visual tasks. Building on these results, we conduct a thorough and detailed analysis to reveal the impact of the *STDChallenge* on trackers.

**Comprehensive and integral experimental analyses (Section 5).** Finally, analysis results from extensive experimental settings show that, in the *STDChallenge*, CPDTrack not only achieved state-of-the-art (SOTA) performance, getting 1.7% superior N-PRE and 1.4% PRE score on *STDChallenge Benchmark* compared to the second algorithm, as shown in Table 1, but also demonstrated remarkable alignment with human DVA, as shown in Fig. 4. Furthermore, it performed well across various tracking benchmarks, particularly excelling in difficult benchmarks [4, 9, 27, 28].

**Contributions.** Our research explores the visual search ability of humans and trackers through detailed analyses based on algorithmic strategies and benchmark evaluations.

## 2 Related Work

**Tracking Benchmark.** In 2013, the VOT competition characterized SOT using five key terms: *single-target*, *model-free*, *causal-trackers*, *single-camera* and *short-term*. The initial three terms serve to differentiate SOT from other visual tasks [29, 30, 31, 32], whereas the last two, *single-camera* and *short-term*, were introduced to simplify early-stage research [6, 7, 19, 21]. Since 2018, several researchers have moved beyond the *short-term* limitation to embrace LTT [33, 9, 34], with the VOT setting a new criterion that tasks permitting complete target disappearance qualify as long-term [10]. More recently, the introduction of GIT [4] has eliminated the *single-camera* restriction, thus offering a more realistic simulation of the world. *STDChallenge Benchmark* is dedicated to examining the distinctions between LTT and GIT compared to STT, and integrates human DVA to steer algorithmic advances.

**Visual Trackers.** Deep learning trackers are typically categorized by motion models into local and global trackers, as shown in Fig. 2. Local trackers, which are most popular method, crop a search region in current frame [11, 12, 13, 15, 35, 16, 36]. These trackers continuously improved by advancements in feature extractor and temporal analysis, have reached SOTA performance on various benchmarks [9, 19, 21]. To tackle *STDChallenge*, it is common to switch to a global detector when local trackers fail, though this transition depends on the local tracker's performance [37, 38, 39, 40, 41]. Local trackers often struggle with proactive global awareness, reducing their robustness against *STDChallenge*. Global trackers, conversely, aim to identify targets throughout the entire image, raising computational demands and easy to be confused by similar objects in the background. Due to these limitations, they are less commonly employed [42, 43]. CPDTrack is designed to model CPD by combining local and global information, thereby enabling precise and efficient global modeling and offering insights for addressing the *STDChallenge*. For a detailed analysis of these trackers, see Appendix D.

**Visual Turing Test.** Traditional evaluation methods, focused on "machine versus machine" comparisons, struggle to comprehensively assess the development of visual intelligence as they rely on predefined metrics in experimental settings. The VTT, at the intersection of turing tests and computer vision, is gaining attention for its "human versus machine" approach, which uses human visual capabilities as a baseline to fully evaluate machine effectiveness [26, 44, 45, 46, 26, 25, 24]. This paradigm has been applied in SOT that simulate human DVA: GIT[4] demonstrates that humans can effectively resolve *STDChallenge*, and PathTracker [22, 47] has developed a circuit model to explain human decision-making processes. We are going to explore the visual search ability of humans and machines when tracking an arbitrary moving target.

## 3 Solving the *STDChallenge*

This section presents the proposed CPDTrack method in detail. We introduce CPD motion model, which divides global information into central and peripheral vision to optimize feature extraction. Additionally, we have developed a temporal clues module that serves as the information query component within CPD theory, facilitating top-down control mechanisms.

### 3.1 CPD Motion Model

A crucial challenge is balancing the maintenance of local resolution and computational efficiency during the extraction of global information. To address this, we incorporate insights from cognitive neuroscience, specifically the fovea concept [48] and CPD, segmenting the current frame into central and peripheral. Their computation is divided into the following steps:

**Determine the position.** For the central vision, as shown in Fig. 3, we use the tracking result $b_{t-1} = (b_{t-1}^{cx}, b_{t-1}^{cy}, w_{t-1}, h_{t-1})$ on the previous frame to determine its position.

**Central vision.** To simulate the decay in visual sensitivity characteristic of the fovea, we employ gaussian distribution model—a method that has found widespread application in vision science [48, 49]. Visual sensitivity is defined as:

$$S(x) = S_0 \cdot exp(-\frac{\left(x - b_{t-1}^{cx}\right)^2}{2 \cdot \sigma^2}),\qquad(1)$$

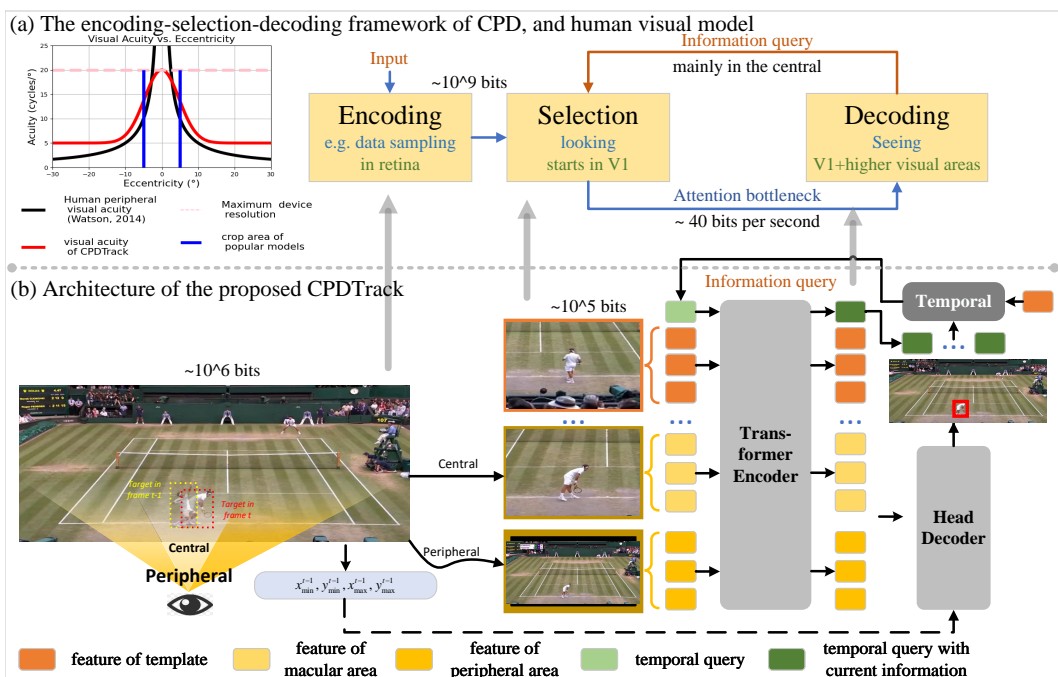

Figure 3: The overall architecture of CPDTrack, referencing the latest one-stream trackers, models CPD. (a) The encoding-selection-decoding framework of CPD. **Visual selection** is the process of choosing the information to focus on, and **visual decoding** is the process of deeply understanding the selected features and information to make cognitive decisions. On the left is a mathematical model of visual selection, where we align human visual information processing with the resolution of devices and machines. (b) Architecture of the proposed CPDTrack: The original frame is treated as encoded visual data. We apply the acuity model described in (a) for visual selecting, which can be modulated by **information query** from temporal. A transformer is employed to replicate the decoding processes occurring in higher brain regions, facilitating complex cognitive tasks. The grey arrows running through both parts highlight the correspondence between the two parts.

where $S(x)$ is sensitivity in $x$ and $S_0$ is the sensitivity in central. we set the $\sigma = \frac{W}{6}$ based on the overall size $(H, W)$ of the image and the 3-sigma rule. This approach ensures that the cumulative visual sensitivity to current frame exceeds 99.7%, effectively covering the vast majority of the visual field.

**Sensitivity of central.** We determine the sensitivity levels within this region using the values from $b_{t-1}$. Integrating sensitivity as:

$$sens_{x-1} = \int_{b_{t-1}^{x_1}}^{b_{t-1}^{x_2}} S(x)dx, \tag{2}$$

where $b_{t-1}^{x_1} = b_{t-1}^{cx} - \frac{w_{t-1}}{2}$ is the top-left coordinate of $b_{t-1}$, and $b_{t-1}^{x_2} = b_{t-1}^{cx} + \frac{w_{t-1}}{2}$ is the bottom-right coordinate of $b_{t-1}$.

**Image crop resizing.** As the central region is resized to a constant size, thus its width $w_{t-1}^e \propto \frac{1}{sens_x}$ and $w_{t-1}^e \propto w_{t-1}$. Therefore, $w_{t-1}^e$ is determined to be:

$$w_{t-1}^e = \frac{w_{t-1}}{sens_{x-1}} = \mathcal{S} \frac{w_{t-1}}{2\Phi(\frac{3w_{t-1}}{W}) - 1}, \tag{3}$$

where $\mathcal{S}$ is a constant factor that can be adjusted, and $\Phi$ is the cumulative density function of the standard normal distribution. Then, we get central region $m_t = (b_{t-1}^{cx}, b_{t-1}^{cy}, w_{t-1}^e, h_{t-1}^e)$.

As the lower limit of the bounding box is set, typically at 10 pixels in tracking, equation (3) will not a division by zero error in practice.

We prove the range $w_{t-1}^e$ in equation (3): First, take the derivative with respect to $w_{t-1}^e(x)$, The range of values for $x$ is $(0, W]$:

$$w_{t-1}^e{}'(x) = \frac{2\Phi(\frac{3x}{W}) - 1 - \frac{6}{W}xf(\frac{3x}{W})}{(2\Phi(\frac{3w_{t-1}}{W}) - 1)^2},$$ (4)

where $f$ is the probability density function of the standard normal distribution. We define the numerator as $n_w(x)$ and denominator as $d_w(x)$

It is clear that $d_w(x) \geq 0$ when $x > 0$; differentiate $n_w(x)$:

$$n_w'(x) = -\frac{18}{W^2}xf'(\frac{3x}{W}).$$ (5)

Clearly, $f' < 0$. Thus, $n_w(x)$ has its minimum value at $x = 0$, $n_w(x) \geq 0$. Therefore, $w_{t-1}^e(x)$is a monotonically increasing function when $x > 0$. When $x$ takes the maximum value, $x = W$, $w_{t-1}^e(x) \approx 0.9987SW$.

Intuitively, the central area is determined by cropping around the target based on the $b_{t-1}$. This differs from traditional motion models, which typically use a static context ratio for cropping. In contrast, the CPD model adapts the contextual range dynamically, offering more flexibility and precision, more visualization in Appendix B.

For the peripheral area, we adjust the entire image to match the size of the central area. This resizing facilitates the simulation of peripheral vision, aligning the scale both regions.

## 3.2 Information query

In CPD [2], the selection process is influenced by cognitive feedback through a top-down approach, as illustrated in Fig. 3. Similarly to mode that humans can actively focus on areas of interest, CPDTrack incorporates an information query mechanism. Spatially, we determine central vision based on the results $b_{t-1}$ of the previous frame. Semantically, we use the query to map high-level semantic information from the previous frame downwards, achieving top-down control. Specifically, each past frame is represented by a token while template is max-pooled into a query as initialization. This query, using cross-attention, extracts refined high-level semantic insights from multiple past frame tokens, then concatenated to images tokens and inputted into the ViT [50]. Within the ViT backbone, the information query interacts with the global information of the current frame and the online template, achieving implicit cognitive feedback's top-down control. Simultaneously, this query integrates the spatio-temporal information of the current frame for use in future frame computations.

## 4  The *STDChallenge Benchmark*

*STDChallenge Benchmark* is an integrated task closely associated with various object tracking subtasks [7, 10, 4]. We select sequences with *STDChallenge* from established benchmarks based on specific sequence level attributes. Furthermore, inspired by cognitive psychology [51], we have designed a dynamic visual capability assessment framework to evaluate and compare the performances of humans and machines when tackling *STDChallenge*. This evaluation directly informs and guides machine development.

**Construct *STDChallenge Benchmark*.** Our objective is not to develop a new dataset to evaluate trackers' visual search ability. Rather, we aim to highlight the often-overlooked issue of *STDChallenge*, already acknowledged by few recent datasets but underemphasized in algorithms design. We have chosen existing benchmarks as our data sources to assess trackers' proficiency with *STDChallenge*, specifically selecting sequences from LaSOT [9], VOTLT2019 [52], and VideoCube [4] where such challenges are existing. Drawing inspiration from cinematography, we introduce the metric

$$STD = \frac{(n_a + n_s) \cdot l_a}{l^2},$$ (6)

to quantify *STDChallenge*, where $n_a$ means the number of *absent*, $n_s$ means the number of *shotcut*, $l_a$ means the length of *absent* and $l$ means the length of the sequence. This formula takes into account the frequency and duration of *absent*, and the frequency of *shotcut*. A higher $STD$ value indicates

greater *STDChallenge*. From LaSOT [9], VOTLT2019 [52], and VideoCube [4], we have compiled a dataset of 252 sequences that constitute the *STDChallenge Benchmark*, detailed attributes and distribution would typically be visualized in Appendix C.

**Visual Turing Test.** Our investigation begins by assessing whether individuals can effectively handle the *STDChallenge*. In line with methodologies from the field of static vision [26, 25], we organize the study using a small-N design [51, 26], enlisting five participants aged 20 to 30. During the experiment, participants watch videos and are instructed to track the target with a mouse cursor. The cursor's position is recorded at each frame by a background program. Sixteen videos (named *STDChallenge-Turing*) sampled from *STDChallenge Benchmark* are displayed in 15 fps, each in a different order across participants. Prior to starting, participants receive detailed training to familiarize themselves with the experimental process and techniques, more details in Appendix E.

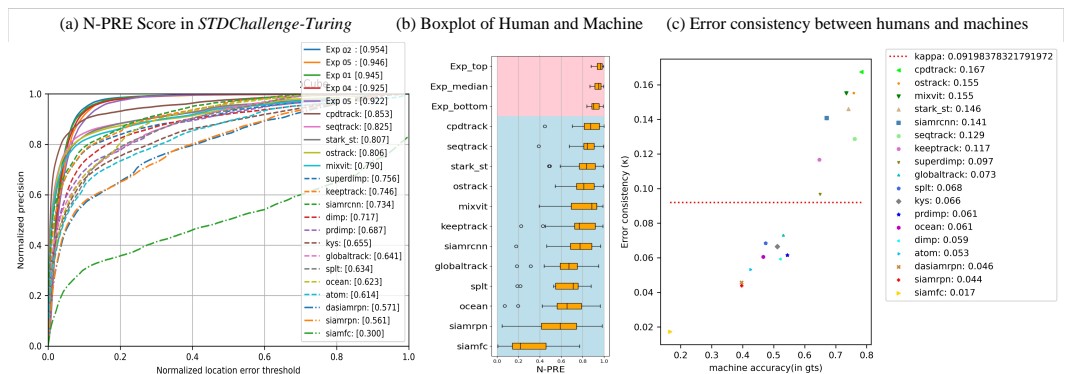

Figure 4: Quantitative indicators show that CPDTrack not only has higher accuracy in the *STDChallenge* but also behaves more similarly to humans. (a) represents the N-PRE score on *STDChallenge-Turing*. (b) represents the distribution of N-PRE scores of humans and machines on various sequences of *STDChallenge-Turing*. (c) represents the error consistency between machines and humans, with kappa representing the average error consistency.

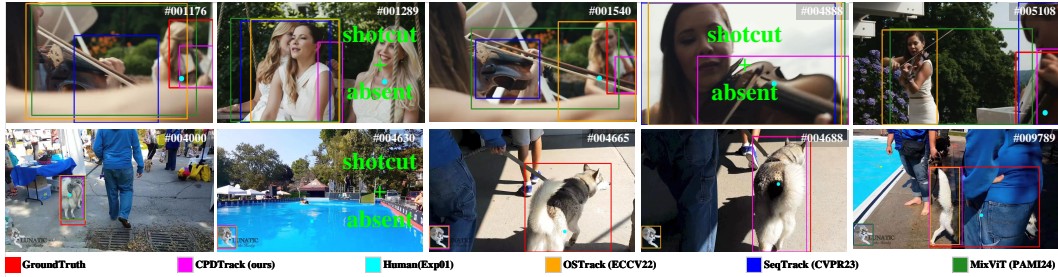

Figure 5: Human results do not necessarily mean correct, but humans can usually quickly re-locate the target after *STDChallenge*. In the upper row, humans can recognize environmental factors closely related to the target in the second image. In the lower row, even if the target is absent, humans are not distracted by the background in the second image. Humans are robust to occlusions in the fifth image.

## 5 Experiments

### 5.1 Visual Turing Test in *STDChallenge-Turing*

*Evaluation metrics.* Given that human typically focuses on a single point and emphasizes an object's salient features rather than its center, we have adjusted our evaluation metrics and outputs accordingly. For error consistency measurements refer to [26], human is successful if the point fall within the groundtruth while trackers is successful if $IoU > 0.5$ in a frame. For performance measurements, we employ the N-PRE metric [4], which evaluates the precision based on relative position between the center of boundingbox and groundtruth, to assess the performance of both humans and machines.

Table 1: State-of-the-art comparison on *STDChallenge*, VideoCube [4] and LaSOT [9]. The top three results are highlight with red, blue and green fonts, respectively.

| Motion Model | Method | *STDChallenge* | | | VideoCube | | | VideoCube R-OPE | | | LaSOT | |
|---|---|---|---|---|---|---|---|---|---|---|---|---|
| | | N-PRE | PRE | SUC | N-PRE | PRE | SUC | N-PRE | SUC | Robust | AUC | P |
| CPD | CPDTrack | 84.2 | 73.3 | 65.9 | 82.9 | 67.1 | 70.4 | 89.5 | 75.6 | 75.3 | 66.1 | 73.0 |
| Local Crop | SeqTrack [13] | 81.9 | 71.9 | 66.8 | 76.8 | 54.0 | 63.5 | 88.3 | 72.5 | 74.6 | 69.9 | 76.3 |
| | OSTrack [12] | 79.1 | 68.9 | 64.6 | 73.7 | 50.7 | 61.8 | 85.8 | 71.3 | 74.4 | 69.1 | 75.2 |
| | MixViT [11] | 82.5 | 71.6 | 66.7 | 76.9 | 52.2 | 63.1 | 88.5 | 72.7 | 74.7 | 69.6 | 75.9 |
| | STARK [53] | 80.7 | 68.2 | 64.5 | 76.3 | 49.4 | 62.1 | 86.8 | 70.4 | 74.5 | 67.1 | - |
| | KeepTrack [15] | 80.4 | 64.3 | 62.8 | 73.0 | 37.9 | 54.3 | 83.0 | 64.4 | 73.8 | 67.1 | 70.2 |
| | Ocean [54] | 57.1 | 39.9 | 40.7 | 53.9 | 19.5 | 34.2 | 74.8 | 51.2 | 73.7 | 56.0 | 56.6 |
| | SuperDiMP [55] | 72.6 | 56.7 | 56.5 | 64.6 | 31.4 | 47.4 | 80.1 | 61.2 | 74.3 | 64.1 | - |
| | PrDiMP [55] | 70.3 | 51.7 | 52.7 | 65.4 | 28.6 | 44.5 | 79.6 | 58.3 | 74.3 | 59.8 | 60.8 |
| | DiMP [56] | 65.9 | 47.0 | 48.6 | 54.6 | 18.7 | 37.1 | 77.2 | 56.0 | 74.0 | 56.9 | 56.7 |
| | SiamRPN [35] | 53.4 | 35.6 | 37.3 | 46.7 | 15.0 | 29.0 | 72.6 | 50.3 | 73.6 | - | - |
| | ATOM [57] | 57.8 | 39.8 | 40.8 | 43.6 | 14.0 | 26.7 | 75.2 | 53.1 | 73.8 | 51.5 | 50.5 |
| | KYS [58] | 60.1 | 42.6 | 44.5 | 49.3 | 17.1 | 33.7 | 80.1 | 59.4 | 73.3 | 55.4 | - |
| | SiamFC [16] | 33.6 | 21.2 | 20.6 | 15.8 | 3.6 | 7.4 | 52.1 | 35.6 | 72.7 | 33.6 | 33.9 |
| Local-Global | SPLT [38] | 60.9 | 38.2 | 40.3 | 56.5 | 15.7 | 33.7 | 72.4 | 47.6 | 73.5 | 39.9 | 42.6 |
| | DaSiamRPN [39] | 53.4 | 35.4 | 37.1 | 46.3 | 14.4 | 29.1 | 72.2 | 50.4 | 73.6 | 42.7 | 44.8 |
| Global | SiamRCNN [43] | 75.3 | 62.8 | 60.7 | 72.6 | 47.9 | 58.8 | 80.5 | 65.8 | 74.5 | 64.8 | - |
| | GlobalTrack [42] | 65.5 | 49.5 | 49.5 | 64.3 | 29.6 | 46.1 | 72.7 | 53.7 | 74.3 | 52.1 | 52.7 |

*Experiments results.* In the *STDChallenge*, humans outperform most trackers, indicating that the DVA of trackers is still far from humans. Furthermore, while the performance of machines tends to vary significantly with $STD$ changing, human consistently exhibit stable tracking ability, as shown in (b) of Fig. 4. Notably, as research progresses, the scores of SOTA trackers are gradually approaching those of humans, suggesting a narrowing gap between machine and human performance. Additionally, the low error consistency between trackers and humans suggest they employ different tracking strategies in past studies. There's a significant positive correlation between the N-PRE of trackers in *STDChallenge* and their error consistency with humans, indicating that trackers are incorporating some implicit human-like strategies to improve performance progressively. Among them, CPDTracker, which explicitly models human strategies, achieved the highest score of $0.167$. This method not only effectively tackles the *STDChallenge* but also aids in guiding machines to exceed the biases of evaluation benchmarks and adapt to real-world scenarios.

*Some Reasoning.* (i) Like global trackers as shown in Fig. 2, humans process entire image information but can handle the *STDChallenge* more effectively at higher frame rates. This demonstrates the superior efficiency of humans in video information processing. (ii) When a target is lost, humans not only quickly become aware of this but also swiftly reacquire the target. This process involves the integrated use of spatial cognition and temporal memory, demonstrating human DVA to understand global information, in contrast to mainstream trackers that rely on local information. (iii) Furthermore, our observations reveal that humans tend to focus on object's salient features over its center when tracking. This significantly leads to current evaluation metrics underestimating human capabilities, while also highlighting the clear target recognition ability of humans.

Moreover, we examined how different modules within the trackers' pipeline influence performance and error consistency. For detailed findings, refer to the Appendix F.

## 5.2 Comparison with SOTA

*STDChallenge Benchmark.* The *STDChallenge Benchmark* is a newly proposed benchmark for LTT that focuses on *STDChallenge*. Designed to refer human visual research ability, it tests capability of trackers to reacquire lost targets. CPDTrack shows improvements of $1.7\%$ *N-PRE* and $1.4\%$ *PRE* over the second-ranked tracker, as shown in Tab. 1. Given the long-tail distributions of *STDChallenge*, we charted performance fluctuations across different $STD$. As illustrated in Fig. 6 (a), CPDTrack excels in robustness and significantly outperforms SOTA trackers, especially in sequences with increasing *STDChallenge* (the last $40\%$, consisting of 100 sequences). We assessed CPDTrack's ability to search targets during the *STDChallenge*. A recovery is deemed successful if IoU between bounding box and groundtruth exceeds 0.5. As illustrated in Fig. 6 (b), CPDTrack reliably maintains target tracking under *STDChallenge*. CPDTrack achieves a recovery success rate of 56.91% within the same time period, significantly outperforming the next best tracker, mixvit.

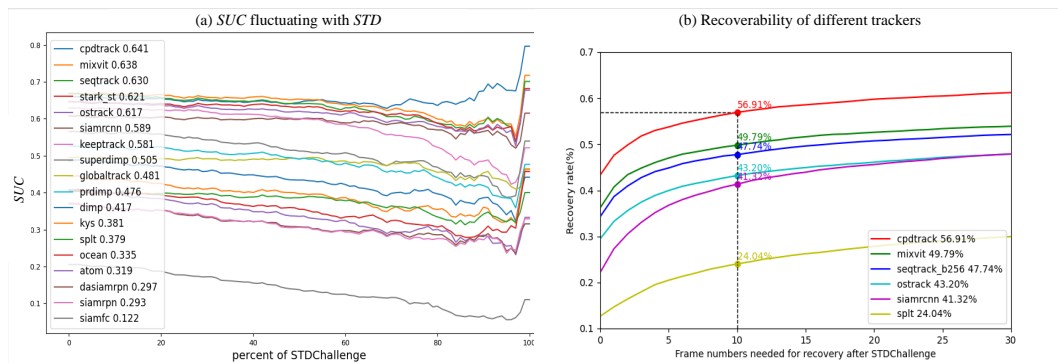

Figure 6: We further emphasize the visual search capabilities of CPDTrack, especially in challenging *STDChallenge* scenarios. (a) performance of *SUC* as the $STD$ changes. We have listed the *SUC* scores on the toughest 100 sequences. (b) represents robustness when facing the *STDChallenge*, where success rate refers to the percentage of frames in which the tracker successfully tracks the target.

Table 2: Ablation studies of CPDTrack on *STDChallenge* and VideoCube. We conducted comparisons of 'central-peripheral' vision and information query.

| # | Method | *STDChallenge Benchmark* | | | | VideoCube | | |
|---|---|---|---|---|---|---|---|---|
| | | consistency | N-PRE | PRE | SUC | N-PRE | PRE | SUC |
| 1 | baseline | 0.159 | 83.8 | 72.9 | 65.6 | 82.7 | 67.0 | 70.2 |
| 2 | central vision | 0.137 | 81.5 | 72.2 | **66.8** | 77.7 | 57.7 | 65.9 |
| 3 | peripheral vision | 0.156 | 80.7 | 69.2 | 61.7 | 80.6 | 60.0 | 65.7 |
| 4 | baseline+query | **0.167** | **84.2** | **73.3** | 65.9 | 82.9 | **67.1** | **70.4** |
| 5 | local as central | 0.165 | 83.0 | 70.6 | 64.5 | **84.3** | 62.9 | 69.0 |

*VideoCube.* VideoCube is a detailed and challenging LTT benchmark aimed at simulating real-world complexities, including View-point changes and disappearances. It features two evaluation methods: One-Pass Evaluation (OPE) and Restarting OPE (R-OPE), the latter allows trackers to restart after failure to evaluate their robustness. CPDTrack outperforms the second-ranked machine across all performance metrics, with improvements ranging from $1.0\%$ to $13.1\%$. Additionally, there has been a notable enhancement in its robustness 75.8, as shown in Tab. 1.

*LaSOT.* LaSOT is a high-quality, large-scale benchmark for LTT featuring a test set of 280 videos with an average of 2448 frames each sequence. We attribute underperformance of CPDTrack to the limited *STDChallenge* scenarios and varying interpretations of groundtruth within these tests as in Fig. 7.

## 5.3 Ablation and Analysis

Through ablation studies, we validate the effectiveness of each module of CPDTrack in *STDChallenge*.

We conducted ablation studies of CPDTrack in *STDChallenge Benchmark* and VideoCube, comparing the impacts of "central-peripheral" vision dichotomy and the use of information query. Information query was removed in #1 as baseline to examine the performance of the CPD Motion Model itself. #2 is traditional local crop, which can be considered as only containing central vision. #3 employed only peripheral vision. #4 is the proposed CPDTrack to validate the effectiveness of cognitive feedback control in ANN. Based on #2, peripheral vision and information query was added in #5 to demonstrate the effectiveness of our designed CPD Motion Model.

**The combination of central and peripheral vision performs better than using either alone.** It can be seen that performance metrics and error consistency of baseline #1 surpass those of #2 using only central vision and #3 using only peripheral vision, with improvements range 0.7% to 9.3% in various metrics on *STDChallenge Benchmark* and VideoCube and more like human, as in Tab. 2. However,

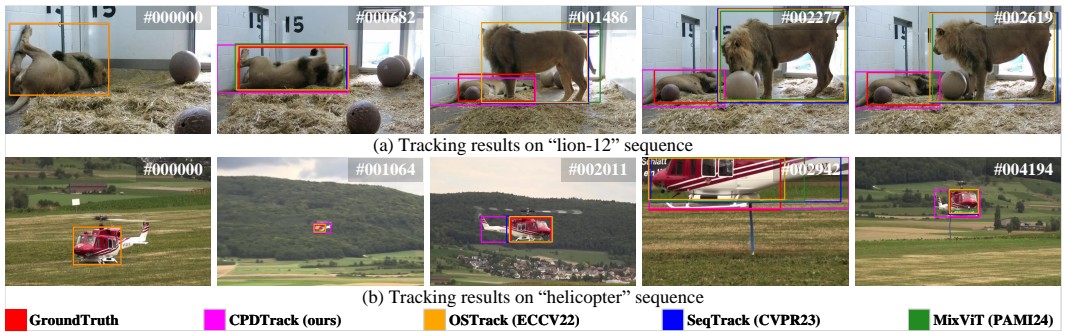

(a) Tracking results on "lion-12" sequence

(b) Tracking results on "helicopter" sequence

■ GroundTruth  ■ CPDTrack (ours)  ■ OSTrack (ECCV22)  ■ SeqTrack (CVPR23)  ■ MixViT (PAMI24)

Figure 7: Visualizations of CPDTrack and some local trackers tracking results. This demonstrates that CPDTrack, influenced by the global perspective, tends to frame the entire target, showing a similarity to human cognition of objects, such as "a lion's tail" or "the tail rotor of a helicopter." This is not an error, but rather a bias introduced by the dataset's setup.

central vision may be better at precise localization. We believe this demonstrates the complementarity of central and peripheral vision, further validating CPD theory.

**Information query is effective.** It can be seen in Tab. 2, compared to baseline#1, #4 has improved performance by 0.1%-0.4% on several metrics in *STDChallenge Benchmark* and VideoCube, and with 0.8% improvement in error consistency. This indicates that in tracking, utilizing higher-level semantic information for top-down control is beneficial for trackers to understand sequence information. On the other hand, information query can also be considered as the retrieval of short-term memory, which suggests that combining short-term and long-term memory (online template) in tracking can better address the *STDChallenge*.

**Modeling central vision with Gaussian acuity is effective.** Compared to the traditional local crop as central vision #5, CPDTrack #4 has surpassed in performance metrics on *STDChallenge Benchmark* and VideoCube, with improvements ranging from 1.2% to 4.2%, as in Tab. 2. The error consistency is also higher than #5. This suggests that considering the bounding box and image size, dynamically extracting context for central vision can better align with peripheral vision.

## 6    Conclusion.

Our study is inspired by the differences in dynamic visual abilities between humans and machines, notably in challenging tasks like LTT and GIT. We attribute this gap primarily to the machines' deficiency in essential visual search skills. Based on CPD from cognitive neuroscience, which categorizes visual inputs into central and peripheral vision, we understand humans' robust visual search ability under limited computational resources.

We propose CPDTrack, extensively validated for its effectiveness and closer behavioral alignment with humans in *STDChallenge Benchmark*. Ablation studies highlights that its success is due to the efficient integration of central and peripheral vision. Our findings indicate that robust visual search capabilities are crucial for gradually adapting trackers to real-world applications.

## Acknowledgments and Disclosure of Funding

This work is jointly supported by the National Science and Technology Major Project (No.2022ZD0116403), the National Natural Science Foundation of China (No.62176255), and the Strategic Priority Research Program of Chinese Academy of Sciences (No.XDA27010201).

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

## Appendices

## A Extended related work.

The attention bottleneck force animals to only process a subset of sensory input in depth. This motivates a unifying central-peripheral dichotomy (CPD), categorizing multisensory processing into central and peripheral senses based on their functions [1, 2, 48, 49, 59, 60, 61].

In the CPD theory, the encoding-selection-decoding framework combined with central and peripheral sensations reflects how the brain processes sensory information in stages to optimize resource usage and cope with environmental challenges.

The encoding stage involves the reception and preliminary processing of all initial sensory inputs entering the sensory system. At this stage, sensory information is converted into signals that the brain can further process.

The selection stage is the phase in CPD where 'central' and 'peripheral' selections are separated. This process is driven not only by saliency mechanisms but also influenced by top-down expectations, which are achieved through higher cognitive processes in the brain.

The decoding stage is the active part of 'central senses' in CPD. After the selection stage, the selected few central sensory inputs are transmitted to higher brain regions for in-depth analysis. This stage requires complex cognitive processing, including integrating information, recognizing patterns, and understanding the deep meaning of inputs. We draw from these principles to construct the CPDTrack.

## B More details of central vision modeling

Using $W = 1024$, we sampled one hundred thousand points from 10 to $W$ and visualized the function.

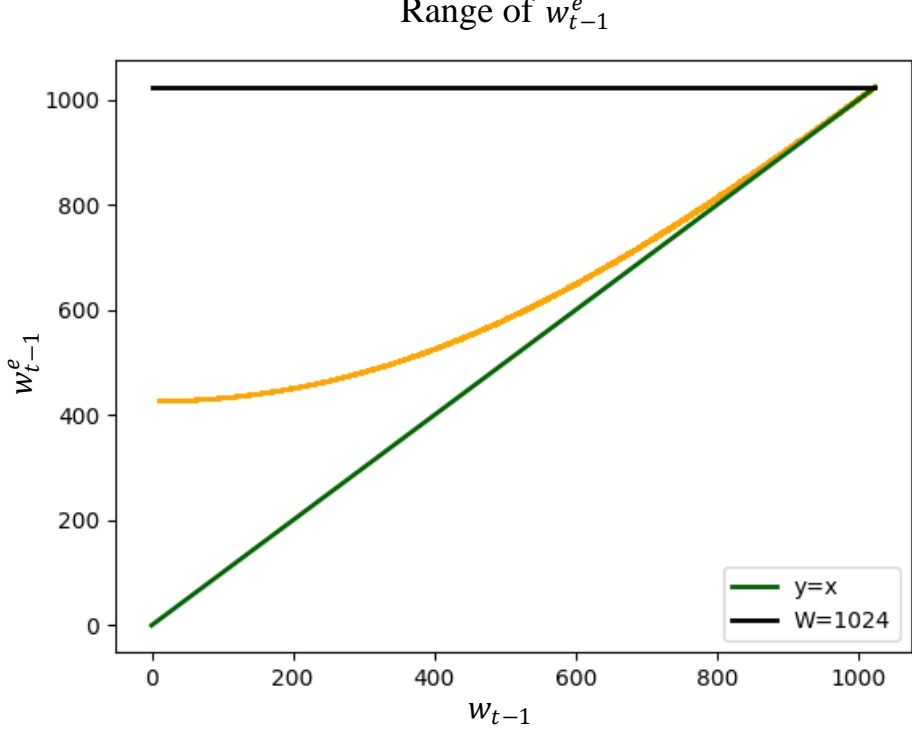

Figure 8: $w_{t-1}^e$ monotonically increases with $w_{t-1}$. The *green* line represents $w_{t-1}^e = w_{t-1}$, while the *black* line represents the image width $W$.

## C Detailed Components of *STDChallenge*

Table 3: Representative Benchmarks in STT, LTT, GIT and *STDChallenge Benchmark*

| Subtask | Benchmark | Videos | Min frame | Mean frame | Max frame | Total frame | *absent* | *shotcut* |
|---|---|---|---|---|---|---|---|---|
| STT | OTB2015[6] | 100 | 71 | 590 | 3872 | 59K | ✗ | ✗ |
| | VOT2016[62] | 60 | 41 | 357 | 1500 | 21K | ✗ | ✗ |
| | VOT2018[63] | 60 | 41 | 356 | 1500 | 21K | ✗ | ✗ |
| | VOT2019[52] | 60 | 41 | 332 | 1500 | 20K | ✗ | ✗ |
| | GOT-10k[19] | 10000 | 29 | 149 | 1418 | 1.45M | ✗ | ✗ |
| LTT | VOTLT2019[52] | 50 | 1389 | 4305 | 29700 | 215K | ✔ | ✗ |
| | LaSOT[9] | 1400 | 1000 | 2502 | 11397 | 3.5M | ✔ | ✗ |
| GIT | VideoCube[4] | 500 | 4008 | 14920 | 29834 | 7.46M | ✔ | ✔ |
| LTT+ GIT | *STDChallenge Benchamrk* | 252 | 1000 | 5192 | 29700 | 1.3M | ✔ | ✔ |

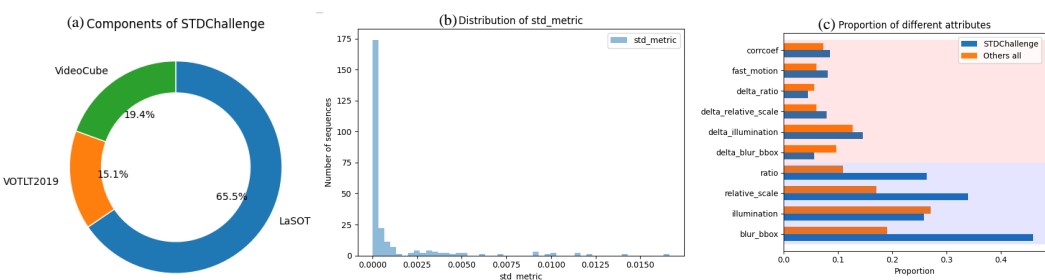

Figure 9: The *STDChallenge Benchmark* is composed of high-difficulty sequences extracted from the LTT and GIT benchmarks. Its $STD$ still exhibits a long-tail distribution.

The *STDChallenge* integrates LTT and GIT tasks to assess the visual search capabilities of tracking machines, which addresses the bias of single datasets, as shown in Fig. 11. This challenge incorporates sequences drawn from the LaSOT [9], VOTLT2019 [52], and VideoCube [4], thereby inheriting their characteristics. The sequence lengths vary, with the shortest being "yoyo-15" from LaSOT and the longest "liverRun" from VOTLT2019. Notably, the extensive dataset of LaSOT contributes significantly to the diversity and complexity of the challenges presented in *STDChallenge*.

*STDChallenge* is a sequence-level challenge in which we analyze the impact of frame-level challenges, as shown in Fig. 10. These frame-level challenges are structured following SOTVerse [18], employing the same computational methods. The *STDChallenge* benchmark is noted for its comprehensive inclusion of various challenges, particularly in terms of *blur_bbox*, *relative_scale*, and *relative_ratio*. Trackers that effectively manage these specific challenges are likely to achieve superior performance within the *STDChallenge* framework.

It can be seen that on the *STDChallenge Benchmark*, the *STD* still shows a long-tail distribution, where the majority of sequences exhibit fewer challenges, while a minority contain a higher number of challenges. There have already been many large-scale benchmarks in target tracking, and how to increase the amount of data under specific conditions is a promising direction.

In our setup, a sequence with *shotcut* or *absent* is deemed to include *STDChallenge*, naturally integrating challenges found in new real-world scenarios. Thus, *STDChallenge* not only incorporates but also enhances other challenging attributes to some extent, as depicted in Fig. 9(c). *absent* is often associated with occlusions and out-of-view scenarios. Formally, *STDChallenge* does not exclude other tasks (such as pathtracker [22]), but we advocate integrating *STDChallenge* with real-world scenarios to avoid ill-posed issues (such as *shotcut* confusing same-looking blocks in pathtracker). Comparison between CPDTrack and pathtracker are in Tab. 4

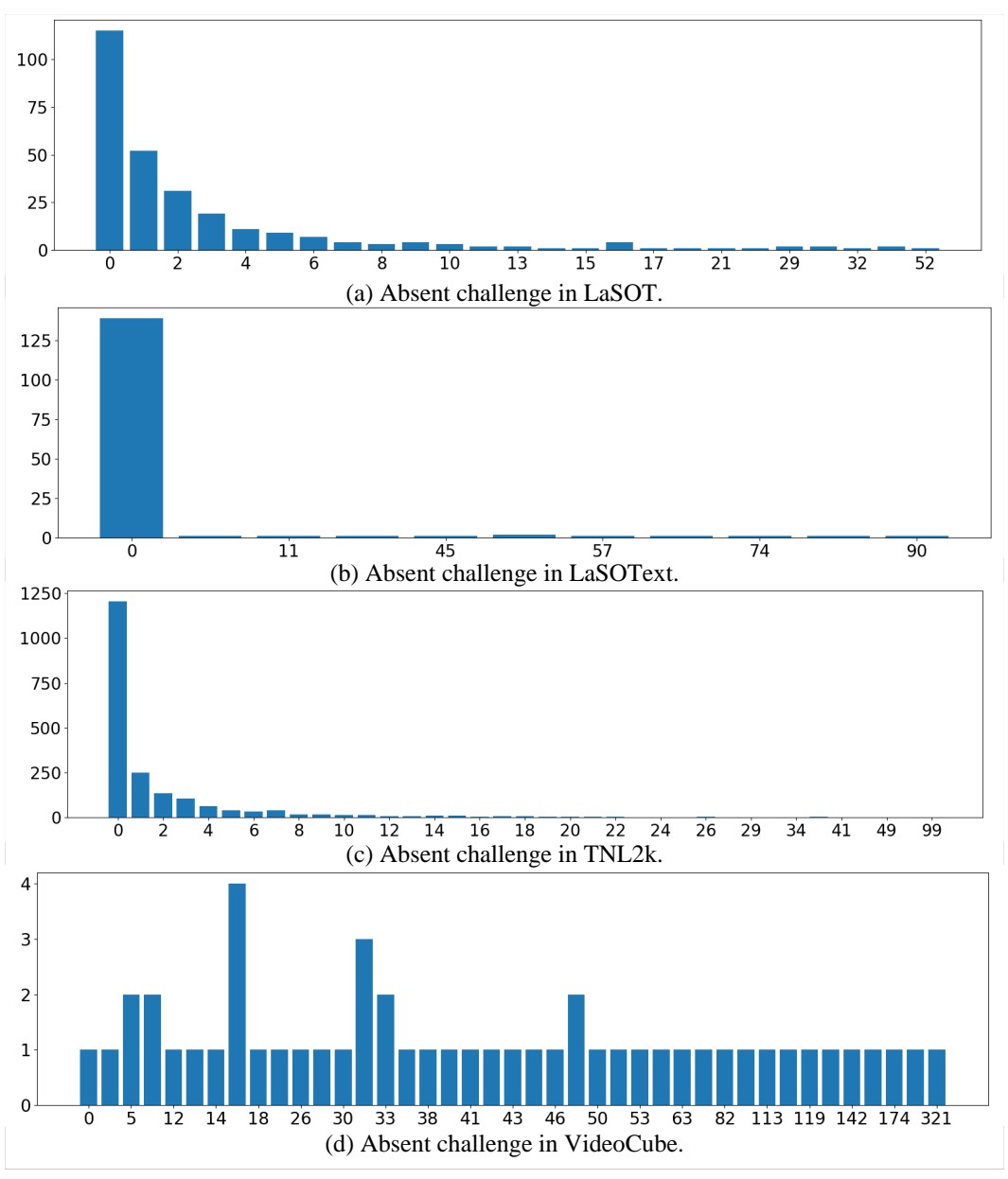

Figure 10: *STDChallenge* exhibits a long-tail distribution across most datasets. The horizontal axis represents the number of *STDChallenge* in each sequence, while the vertical axis represents the number of sequences.

Table 4: From the model design perspective, we believe TransT+InT aims to address PathTracker, a challenge that relies on trajectory information rather than appearance features, which diverges from the Motivation of *STDChallenge*. We attempted to replicate TransT+InT in VideoCube and *STDChallenge*; however, we did not achieve optimal performance due to the lack of clear documentation as the issue in the github. Therefore, we believe these can partly explain the suboptimal performance of TransT+InT on *STDChallenge*.

| Motion Model | Method | Error Consistency | *STDChallenge* | | | VideoCube | | |
|---|---|---|---|---|---|---|---|---|
| | | | N-PRE | PRE | SUC | N-PRE | PRE | SUC |
| CPD | CPDTrack | 16.7 | 84.2 | 73.3 | 65.9 | 82.9 | 67.1 | 70.4 |
| Local Crop | TransT | 13.9 | 73.9 | 62.2 | 59.1 | 69.2 | 42.9 | 55.1 |
| | TransT+InT | 9.2 | 70.6 | 59.0 | 56.4 | 61.0 | 36.7 | 47.0 |

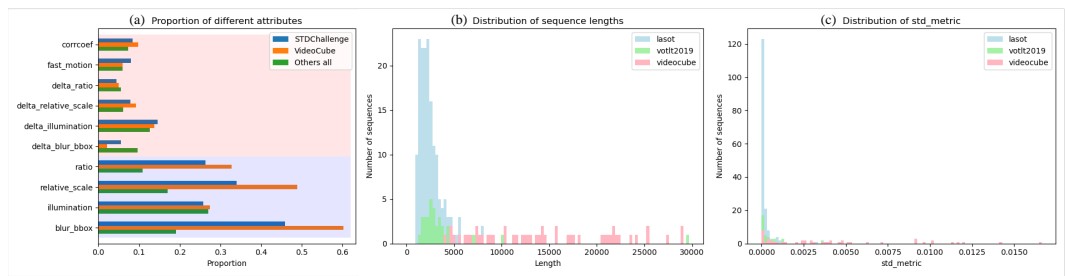

Figure 11: The *STDChallenge Benchmark* is composed of multiple datasets and has a more rational distribution of challenging attributes. Compared to VideoCube, it expands the range of *STD* and lengths of sequence, mitigating biases in single dataset and thus providing a more comprehensive assessment environment. It is recommended to view this in enlarged and color format for better clarity.

# D    Machines.

## D.1    Implementation Details

*Model.* We adopt ViT-B[50] as our encoder architecture for CPDTrack. The encoders are initialized with the MAE[64] pre-trained parameters. The patch size is set to $16 \times 16$. The decoder consists of 2 transformer blocks. The decoder hidden size is $256$, the number of attention heads is $8$, and the hidden size of the feed forward network (FFN) is $1024$. The number of quantization nbins and the vocabulary size are all set to $4000$. The dimension of word embedding is $256$, which is consistent with the decoder hidden size. The embedding-to-word sampling uses a 3-layer perceptron followed by a softmax. The hidden dimension of the perceptron is $256$. The output dimension is nbins, which is aligned with the number of words in vocabulary . The word with the maximum likelihood is sampled as the output word. In addition, we present model parameters, FLOPs, and inference speed in Tab.5.

*Training.* Our training data includes the training splits of VideoCube[], LaSOT[9], GOT-10k[19], and TrackingNet[21]. Aligned with VOT2020 evaluation protocol[65], we remove the 1k forbidden videos in GOT-10k during training. For the evaluation on GOT-10k test set, we follow the official requirements and only use the training split of GOT-10k. Brightness jittering are used for data augmentation. We train the model with AdamW[66] optimizer and set the learning rate of the encoder to $1e-5$, the decoder and remaining modules to $1e-4$, and the weight decay to $1e-4$. The model is trained for a total of 300 epochs with 60k image pairs per epoch. The learning rate decreases by a factor of 10 after 240 epochs. The model is trained on a server with four A5000 GPUs and is tested on an A5000 GPU. The tracking speed is about 23 FPS

*Inference.* We use the first template and online template together with the central region and the peripheral region as input of CPDTrack. The online template update interval T is set to $400$ by default, while the update threshold $\tau$ is set to $0.7$. CPDTrack will attempt to update online template within $10$ frames; if none meet the threshold, it will not update.

## D.2    Local Crop Trackers

**OSTrack [12].** A novel single-stream tracking framework has been proposed, which unifies feature learning and relationship modeling. At the same time, an candidate prior elimination module has been introduced to eliminate background interference while improving computational efficiency.

Table 5: Details of CPDTrack.

| Model | Encoder | Params (M) | FLOPs (G) | Speed (fps) |
|---|---|---|---|---|
| CPDTrack | ViT-B | 119 | 88 | 23 |

**MixViT [11].** MixViT proposes a Mixed Attention Module (MAM), used for simultaneously extracting features and integrating target information. This synchronous modeling approach can extract distinctive features of specific targets and facilitates extensive communication between the target and the search area. In this version, ViT is used to compute attention.

**SeqTrack [13].** The predictive object bounding boxes are generated using an autoregressive method, which simplifies the loss function.

**STARK [53].** Using an encoder-decoder for relational modeling, and a head that directly predicts the corners of bounding boxes has been proposed.

**KeepTrack [15].** KeepTrack combines SuperDiMP with a target candidate association network to achieve robust tracking. The authors retrained the target candidate association network on challenging sequences extracted from LaSOT.

**Ocean [54].** Ocean also adopts an anchor-free structure. The backbone network uses parameters pre-trained on ImageNet for initialization.

**PrDiMP and SuperDiMP [55].** PrDiMP and SuperDiMP use probabilistic regression to improve accuracy.

**DiMP [56].** Based on the framework proposed by ATOM, DiMP optimizes the loss function to achieve stronger discriminative ability. The backbone network is initialized with ImageNet weights.

**SiamRPN [35].** SiamRPN introduces a region proposal network to achieve precise object regression.

**ATOM [57].** ATOM attempts to combine CF and SNN and proposes a new framework to take advantage of offline training and online updates.

**KYS [58].** KYS represents scene information as state vectors and combines them with the appearance model to locate objects.

**SiamFC [16].** As the pioneer of SNN-based trackers, SiamFC achieves satisfactory tracking performance by matching features between the template area and the search area through a simple network structure.

### D.3 Local-Global Trackers

**SPLT [38].** SPLT designed a verifier to switch between global search and local search. SPLT uses it as a feature extractor and down-samples the spatial resolution of template features to 1X1 through average pooling. In terms of the verification model, SPLT employs ResNet50 as the backbone of the verifier.

**Learning regression and verification networks for robust long-term tracking [41].** It employed a verification network with local view to identify the target from the detected candidates. If the target disappears, the learning based switching scheme determines whether to trigger the global search mode.

**Effective Local and Global Search for Fast Long-Term Tracking [40].** It used the score from a target verifier within the local search module to decide whether to switch strategies.

**DaSiamRPN [39].** DaSiamRPN uses data augmentation to enhance its discriminative ability.

**RTracker [37].** RTracker uses a tree-structured memory to dynamically link the tracker and detector by controlling the collection of positive and negative target samples, thereby enabling self-recovery capabilities.

### D.4 Global Trackers

**SiamRCNN [43].** SiamRCNN utilizes a re-detection mechanism and a small trajectory dynamic programming machine to address the issue of object disappearance. SiamRCNN is based on the implementation of FasterRCNN, using ResNet-101-FPN as its backbone.

**GlobalTrack [42].** GlobalTrack does not assume motion consistency and performs a full image search to eliminate cumulative errors.

# E Experiment Organization.

This section provides a comprehensive overview of the experimental setup for the VTT, including composition of test data, selection of equipment, experimental instructions, experimental procedures, survey questionnaire, and grouping information.

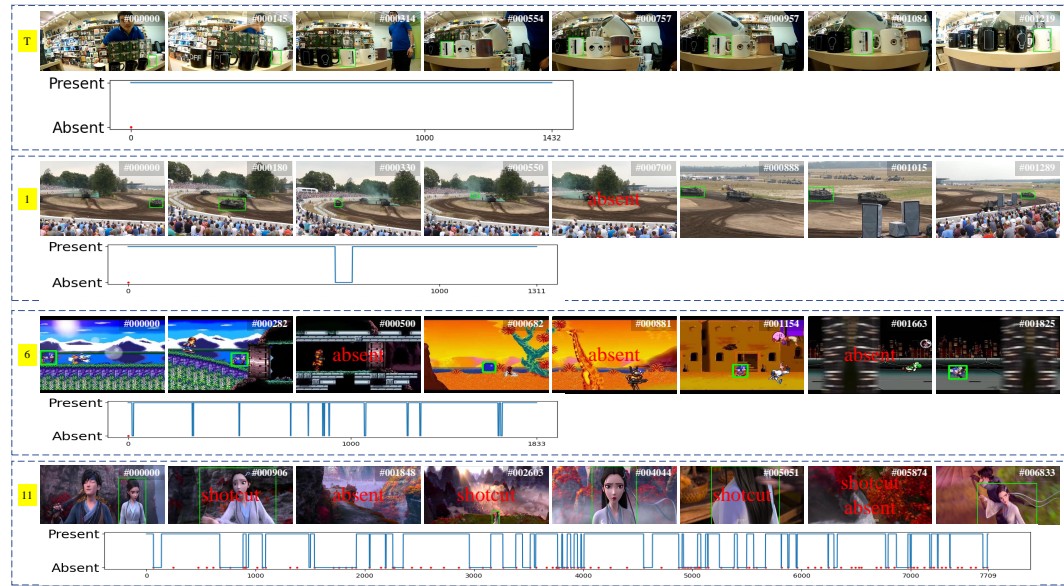

Figure 12: Representative sequences used in Visual Turing Test: (T) TEST video example, which helps participants familiarize themselves with the operational process. (1) First group example from the STDChallenge Benchmark. Lower difficulty, containing fewer *absent*, almost no *shotcut*, and no changes in scenery. (2) Second group example from the STDChallenge Benchmark. Medium difficulty, containing some *absent*, fewer *shotcut*, and some changes in scenery. (3) Third group example from the STDChallenge Benchmark. Higher difficulty, containing many *absent*, more *shotcut*, and significant changes in scenery.

## E.1 Experiment Environment

In our dataset preparation, we initially categorize the STDChallenge benchmark into three distinct groups based on the $STD$ as previously described: sequences with $STD < 10^{-4}$, sequences with $10^{-4} < STD < 10^{-3}$, and sequences with $STD > 10^{-3}$. From each category, five sequences are randomly selected and paired with a designated test sequence to establish the *STDChallenge-Turing* test environment. This environment is composed of 16 sequences that represent a spectrum of challenges within *STDChallenge Benchmark*, providing a robust framework for evaluating performance across varying difficulty levels.

## E.2 Experiment Statement

We have communicated with our institution's IRB, submitted the necessary materials, and obtained IRB approval. The experimental tasks include watching videos and moving the mouse to track a target. To avoid the risk of worsening the participants' vision conditions, we allow participants to rest between two videos. Additionally, participants are allowed to adjust their chairs and the distance to the screen to achieve the most comfortable conditions. Participants will be compensated 100 RMB per hour. The total amount used is 1000 RMB.

Participants were asked to review and sign the following declaration of consent:

**Psychophysical research.** You will be asked to watch video sequences on a computer screen and track a target using a mouse. For completing a session, you will be compensated at a rate of 100 RMB per hour.

**Declaration of consent.** I agree to participate in a behavioral experiment on visual perception. I participate voluntarily in this study. I have been informed that I can stop the experiment at any time without giving any reason and without any negative consequences. I know that I can contact the experimenters at any time to ask questions about the research project.

**Declaration of consent for data processing and data publication.** I consent that the experimental data obtained during the experiment may be used for scientific evaluation and publication in a semi-anonymous form. I agree to have my personal data (such as name, phone number, address) stored digitally; except for being used to contact me, it may not be used for any other purposes. These personal data will only be retained within the researchers' group and will never be transferred to third parties.

### E.3 Statistical Testing

Referencing [25], we focused on ensuring the accuracy and validity of data collected from a limited number of subjects to accurately reflect human visual capabilities. Consequently, we implemented measures to prevent out-of-distribution (OOD) responses, enabling the use of all responses. 1) We did not use crowdsourcing platforms; instead, the Visual Turing Test were conducted in a laboratory environment under supervision. 2) Before the experiment, we assessed participants' cognitive and perceptual levels to ensure normalcy, and allowed them to familiarize with the equipment through basic exercises. 3) During the experiment, participants were permitted to pause the experiment a certain number of times voluntarily to adjust their state. 4) In cases where the mouse cursor moved off-screen, we retained the results from the previous frame until the cursor returned within the screen area.

### E.4 Questionnaire.

*Q1. Your name:*

*Q2. Your id:*

*Q3. Your age:*

*Q4. Your vision condition:*

*Q5: Approximately how many times were you unable to locate the target?*

*Q6: How do you feel about the video playback speed?*

*Q7: Do you feel that the videos vary in difficulty, and which ones do you find more challenging?*

### E.5 Device Selection

Measuring human DVA is a crucial aspect of our research. Traditionally, studies have employed eye trackers or computer mice to assess visual capabilities. Our preference for using a mouse over an eye tracker is based on several considerations:

**(1)** Eye trackers, which passively record eye movements, are influenced by multiple factors including device accuracy, user posture, the distance between the eye and the device, and the presence of eyeglasses, leading to potential limitations in data accuracy.

**(2)** Eye trackers have greater cumulative errors. When errors occur, subjects can only adjust their posture and cannot actively correct the capture results. However, when significant cumulative errors occur with the mouse, we allow subjects to actively pause the video and correct the errors.

**(3)** Some researchers have also pointed out from a theoretical perspective that eye trackers are affected by the tracking rate (*i.e.*, the amount of eye movement data lost). Blinking or brief shifts in gaze can result in a lower tracking rate, which requires subjects to remain highly focused while using the eye tracker. However, the video sequences we test are long, inevitably affecting the performance of the eye tracker due to the subject's fatigue.

**(4)** The use of a mouse to document observation points is well-established within both academic and industrial settings, providing a reliable and practical alternative for tracking visual observation in long-duration tasks.

## E.6 Experiments Process

Process of DVA Experiments:

**(1)** Participants initially inspect the setup equipment and adjust their chair height, seating posture, and their distance from the screen to ensure comfort and optimal viewing conditions.

**(2)** A test video then plays centrally on the screen. In the first frame, participants are tasked with observing and memorizing specific features of a designated target, subsequently using the mouse to track this target throughout the remaining video frames.

**(3)** Upon completion of the test, the lead researcher addresses any questions from participants, clarifying any uncertainties.

**(4)** A series of 15 official videos from varied groups are sequentially displayed, with breaks between videos to reduce visual strain. During these sessions, participants are required to diligently track the target and accurately maintain the mouse's position.

**(5)** Finally, participants complete a self-assessment questionnaire to evaluate their own performance and experience during the test.

## E.7 Group Information

Table 6: Subjects and corresponding videos. Everyone must watch the same videos, but in different orders. One of the sequences is the TEST sequence, used to help subjects familiarize themselves with the process.

| index | sequence | STD | length | Exp_01 | Exp_02 | Exp_03 | Exp_04 | Exp_05 |
|-------|----------|-----|--------|--------|--------|--------|--------|--------|
| *TEST* | cup-1 | 0.0 | 1433 | *TEST* | *TEST* | *TEST* | *TEST* | *TEST* |
| *01* | kangaroo-5 | $4.996 \times 10^{-5}$ | 3119 | *13* | 01 | *02* | *14* | 02 |
| *02* | tank-9 | $3.195 \times 10^{-5}$ | 1312 | *05* | 02 | *03* | *15* | 04 |
| *03* | racing-15 | $4.279 \times 10^{-5}$ | 1070 | *15* | 03 | *10* | *10* | 08 |
| *04* | mouse-17 | $5.105 \times 10^{-5}$ | 1608 | *09* | 04 | *07* | *05* | 13 |
| *05* | sepia-8 | $3.363 \times 10^{-5}$ | 3694 | *02* | 05 | *04* | *13* | 15 |
| *06* | 362 | $2.807 \times 10^{-4}$ | 12704 | *06* | 06 | *01* | *11* | 06 |
| *07* | squirrel-13 | $2.190 \times 10^{-4}$ | 2000 | *12* | 07 | *09* | *03* | 05 |
| *08* | gametarget-2 | $2.473 \times 10^{-4}$ | 1834 | *10* | 08 | *06* | *09* | 14 |
| *09* | 436 | $4.611 \times 10^{-4}$ | 11531 | *01* | 09 | *08* | *07* | 12 |
| *10* | bottle-12 | $2.361 \times 10^{-4}$ | 2400 | *07* | 10 | *05* | *08* | 10 |
| *11* | 015 | $6.290 \times 10^{-3}$ | 7710 | *03* | 11 | *15* | *02* | 03 |
| *12* | 373 | $4.037 \times 10^{-3}$ | 12972 | *08* | 12 | *14* | *12* | 07 |
| *13* | 401 | $4.860 \times 10^{-3}$ | 4978 | *14* | 13 | *11* | *06* | 09 |
| *14* | 350 | $1.159 \times 10^{-3}$ | 6648 | *11* | 14 | *13* | *01* | 11 |
| *15* | 033 | $4.074 \times 10^{-3}$ | 14300 | *04* | 15 | *12* | *04* | 01 |
| length | | | 89313 | 89313 | 89313 | 89313 | 89313 | 89313 |

# F  Performance of Human and Machines

## F.1  Error Consistency of Humans and Machines

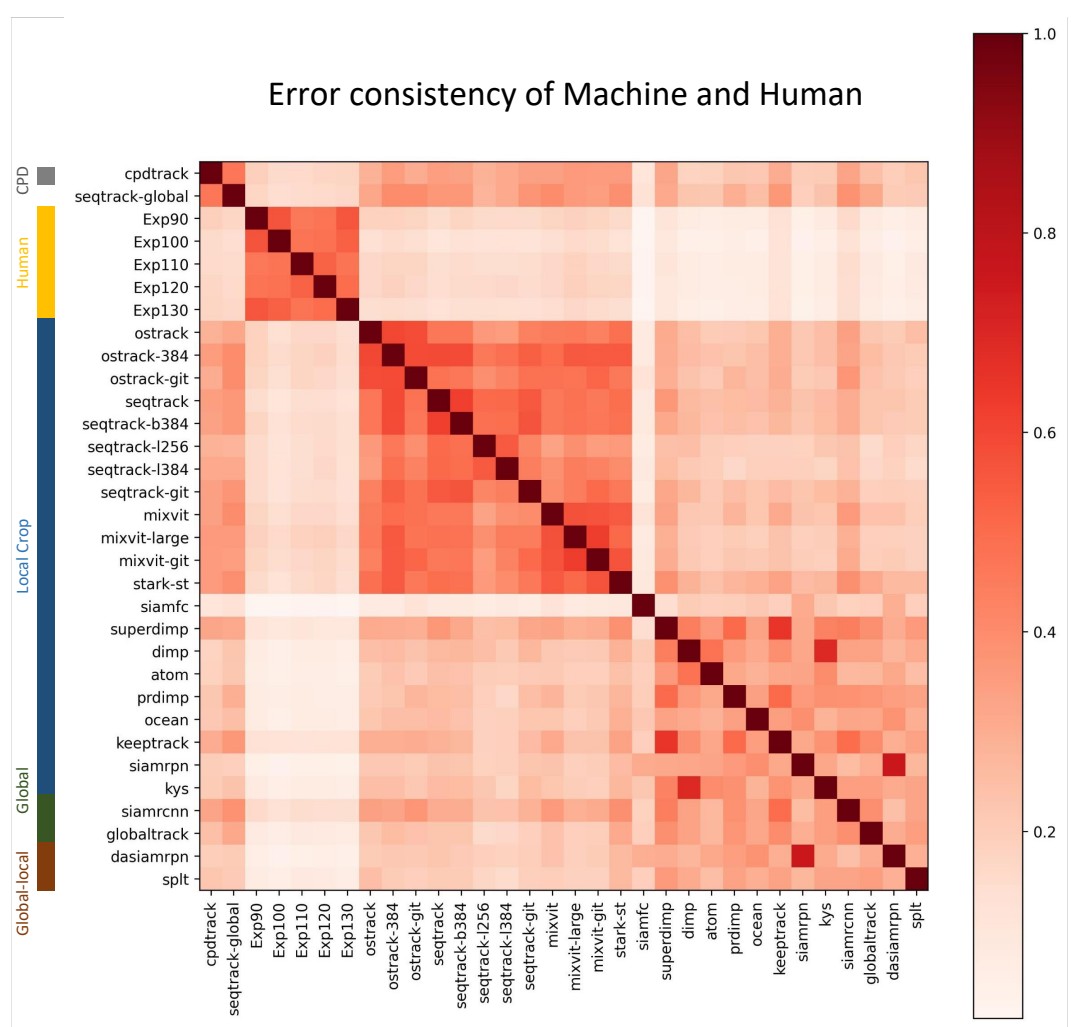

Figure 13: The error consistency differs among decision-makers (machines/humans) in *STDChallenge*. This represents the degree of behavioral similarity between different decision-makers. For each sequence in the *STDChallenge-Turing Benchmark*, we calculate the error consistency o each frame for different decision-makers, and then average these across all sequences to obtain the overall error consistency among the decision-makers.

As shown in Fig. 13, CPDTrack exhibits a higher degree of consistency with human behavior compared to other machines, yet it remains closely aligned with machines. Notably, it demonstrates a higher level of consistency with trackers that utilize ViT as their backbone. This observation underscores the significant influence that backbone can have on algorithmic performance. Furthermore, CPDTrack shares more similarities with trackers that implement a global strategy, under the same network architecture. We attribute this similarity to the adoption of strategies that more closely mirror human cognitive processes.

As shown in Fig. 13,ViT-based trackers display high similarity due to shared components like the backbone and motion model. Enhanced parameters, increased resolution, and more training data have improved their alignment with human error consistency. This is believed to result from the labels in supervised learning being annotated by humans, enhancing the algorithms' ability to fit these labels.

As shown in Fig. 13, compared to ViT-based trackers, those based on CNN exhibit lower similarity among themselves. Nonetheless, certain algorithms that build on others still achieve high consistency in error consistency.

As shown in Fig. 13, the motion model plays a crucial role in this consistency, particularly affecting how trackers process and interpret video sequences, a difference that becomes particularly pronounced in high-resolution settings.

## F.2 Effect of different modules in pipeline on trackers

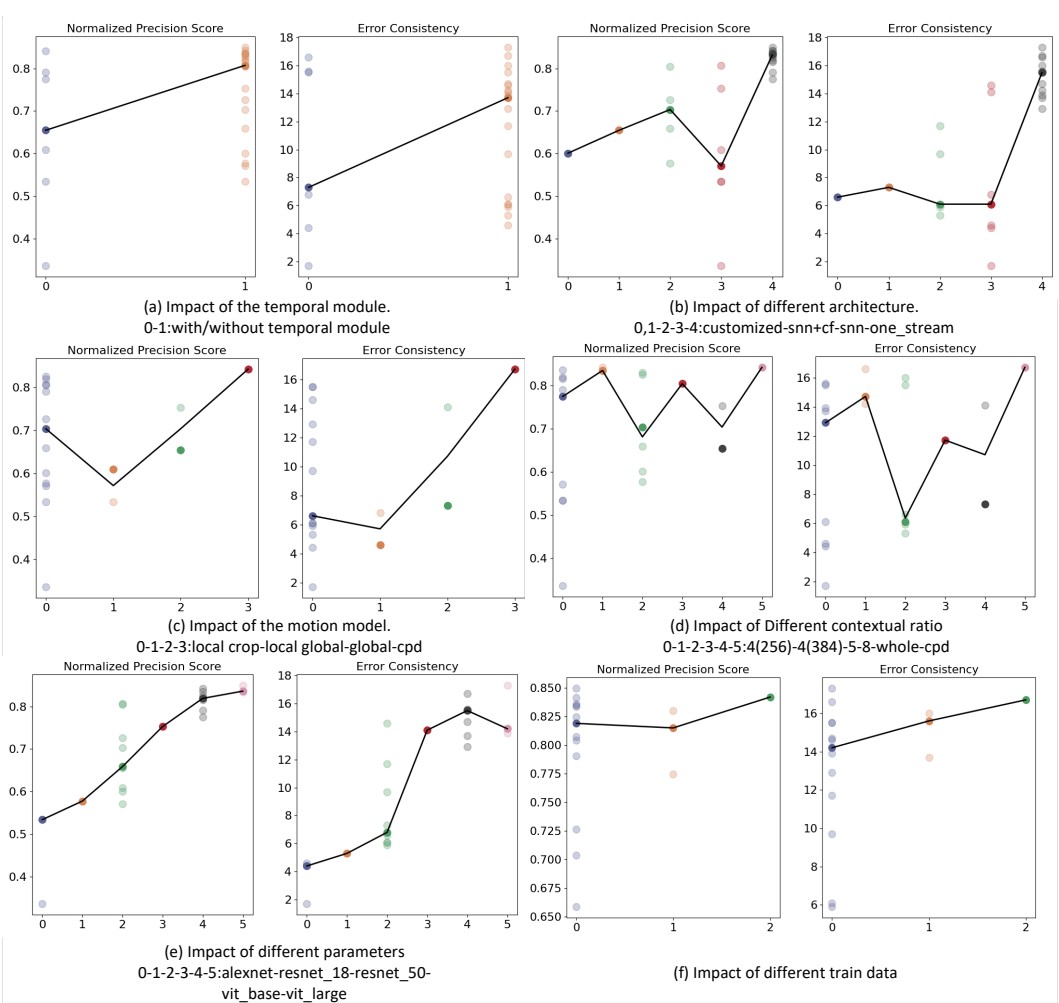

Figure 14: Impact of different settings in tracker pipeline. We highlight the median in each setting. In (f), "1" means the training set contains GOT-10k, COCO, TrackingNet, and LaSOT, while "2" means the training set contains GOT-10k, COCO, TrackingNet, LaSOT, and VideoCube, "3" means the training set contains GOT-10k, TrackingNet, LaSOT, and VideoCube.

(a) As shown in Fig. 14(a). Using temporal information is more effective and also more human-like.

(b) As shown in Fig. 14(b). Compared to other network architectures, one-stream is more effective and also more human-like. However, since current one-stream trackers are all based on ViT, it is unclear whether the effects are due to one-stream or ViT.

(c) As shown in Fig. 14(c). Undoubtedly, the CPD motion model has better effects and more human-like behavior. At the same time, compared to the local-based motion model, the global one is more human-like, as it treats the entire image as central vision, improving performance at the cost of computational efficiency.

(d) As shown in Fig. 14(d). Increasing the context ratio does not seem to change the performance and error consistency, but after changing the resolution from 4(256) to 4(384), there has been an improvement. How to maintain computational efficiency while increasing resolution is a promising direction.

(e) As shown in Fig. 14(e). Obviously, increasing the number of parameters and switching to a newer backbone can significantly improve performance and error consistency. We believe this is due to the increased parameters enhancing the network's ability to fit human labels. The error consistency of vit_large has decreased, and we believe this might be because the network has learned some strategies that surpass human capabilities.

(f) As shown in Fig. 14(f). Increasing training data does not necessarily mean an improvement in performance, but it often enhances error consistency with humans. We believe this is because richer human labels enhance the network's learning capabilities.

# G  Broader Impact and Limitation

**Broader Impact.** The fundamental objective of our research is to model human visual search capabilities through CPD theory. The *STDChallenge Benchmark* facilitates our study by aligning human and machine performances in a complex visual task designed to explore tracking strategies under spatio-temporal discontinuity. Our developed CPD Motion Model has not only achieved SOTA results in this task but has also enhanced the behavioral similarities between humans and machines, substantiating CPD theory and illuminating the potent visual search abilities of humans under constraints of limited information. This work underscores the significance of human-like modeling in enhancing tracking algorithms.

However, we must be cautious about the potential misuse of this technology, particularly in surveillance applications, which could have serious ethical implications. Similarly, the development of highly human-like algorithms has already sparked some concerns, similar to those associated with large language models (LLMs). Despite these challenges, the technology holds crucial potential for beneficial applications in fields such as sports, autonomous driving, robotics, and broader machine vision contexts in the real world. To promote responsible usage and further research into these positive applications, we plan to make our code and data publicly available.

**Limitation.** CPDTrack focuses on the STDChallenge but neglects performance in simpler scenarios, leading to its underperformance on some datasets compared to SOTA trackers, as shown in Tab. 7. In the future, we plan to further improve it to adapt to various environments. The human-like modeling approach adds overhead, making CPDTrack less immediately applicable to real-world applications as it currently stands. *STDChallenge* in the *STDChallenge Benchmark* still shows a long-tail distribution, meaning most sequences contain fewer *STDChallenge*. This is still far from the real world, and how to increase data under specific challenges remains an open question.

**Generalization of CPDTrack.** We believe that many simple scenarios in Single Object Tracking (SOT) are based on laboratory settings and even possess characteristics of toy scenarios (for example, tracking smoothly moving targets within very short time) to some extent. In contrast, CPDTrack extends much further into the real world:

Local trackers introduce stronger priors, which enhance their performance in STT scenarios. Therefore, *STDChallenge* is fatal for them [18, 17]; however, CPDTrack effectively addresses this challenge.

Setup of the *STDChallenge Benchmark* is that any sequence containing at least one *absent* or *shotcut* is considered to include the *STDChallenge*. It is evident that the *STDChallenge Benchmark* already includes many relatively simple video environments, as shown in Fig. 12 (1).

"Experiments indicate that humans perform worse than SOTA trackers in simple scenes, likely due to the precise focus of local trackers. While a global perspective, used by humans and CPDTrack, offers a comprehensive view, it may not align with the dataset setup (e.g., 'lion's tail' or 'helicopter tail rotor'; see Fig. 7)."

However, in complex scenarios, human capabilities to track moving targets surpass those of trackers, a trait also inherited to some extent by the human-like modeling of CPDTrack.

**Computation overhead of CPDTrack.** We acknowledge that compared to some mainstream trackers, CPDTrack's human-like modeling causes some additional computational costs. However, in our experimental setup, CPDTrack achieves 20-30 fps, which already meets the requirements for real-time performance.

Moreover, the main contribution of our paper is to validate the effectiveness of human-like modeling through CPDTrack, and the experimental conclusions have affirmed the motivation behind our work. While there are computational costs, these do not overshadow the contributions of CPDTrack. We greatly appreciate your questions and will continue to explore improvements in future work.

Table 7: Comparison with state-of-the-art methods on additional benchmarks, where is AUC scores on TNL2k and LaSOText, and AO scores on GOT-10k. We add a symbol * over GOT-10k to indicate that the corresponding models are only trained with the GOT-10k training set.

| | SiamFC[16] | ECO[67] | Ocean[54] | ATOM[57] | DiMP[56] | TransT[14] | OSTrack[12] | SeqTrack[13] | CPDTrack |
|---|---|---|---|---|---|---|---|---|---|
| TNL2k[20] | 29.5 | 32.6 | 38.4 | 40.1 | 44.7 | 50.7 | 54.3 | 54.9 | 50.4 |
| LaSOText[9] | - | - | - | 37.6 | 39.2 | - | 47.4 | 49.5 | 42.0 |
| GOT-10k*[19] | 34.8 | 31.6 | 61.1 | 55.6 | 61.1 | 67.1 | 71.0 | 74.7 | 63.7 |

