# OpenReview forum: "Beyond Accuracy: Tracking more like Human via Visual Search"
_NeurIPS.cc/2024/Conference — NeurIPS 2024 poster_

### Official Review · Reviewer_16gz · 2024-07-07

**Soundness:** 3
**Presentation:** 3
**Contribution:** 4
**Rating:** 7
**Confidence:** 5

**Summary:**

The authors touch upon a very important topic in visual tracking. They try to build upon the Central-Peripheral Dichotomy (CPD) theory that talks about how humans use visual information to track targets in complex environments. To this end, they propose a tracker named CPDTrack and STDChallenge Benchmark (for spatio-temporal discontinuities). They claim that their CPDTrack achieves SOTA performance on STDChallenge, aligns with human behavior and generalizes across various benchmarks.

**Strengths:**

The authors clearly highlight the importance of aligning visual tracking in both humans and machines, and to that respect also conduct a human study.
They also cover the related works decently well to highlight the previous works in the field.
The distinction they made between LTT and GIT, and the attempt at doing a Visual Turing Test makes the paper stand out.
The absolute results presented on the STDChallenge also show performance gains compared to the other networks (albeit small).

**Weaknesses:**

The authors do not have IRB which is one of the basic requirements for working with human subjects.
I also did not see any strong statistical testing for human vs machine responses. Seems like to authors selected responses from all human participants. If not, i don’t see any methods for dropping the out of distribution responses.
The authors also conducted 5 experiments with human subjects, but i didn’t find a clear link between the selected experiments for humans and the STDChallenge benchmark.
There is also no justification for why removing information query was treated as baseline in ablation studies. Please justify.
For the STDChallenge benchmark, i didn’t find a justification as to why only the said challenges were included in the benchmark. The authors should clarify that if possible.
How easy/hard is it to include newer challenges in the benchmark?

**Questions:**

There is another similar algorithm called DorsalNet. Have the authors considered the similarity/differences with it?
For the citation of PathTracker, the paper cited is InT which is a solution to the PathTracker challenge. For completeness, the challenge was introduced in https://arxiv.org/abs/2110.02772 which the authors should consider citing.
In fig 5 the authors say that both CPDTrack and humans are robust to occlusions, environment changes and absence of target. InT has similar claims; have the authors tested their benchmark with the TransT+InT proposed in the paper ``Tracking without re-recognition”?

**Limitations:**

The gains are small compared to other networks on the STDChallenge benchmark. Have authors made an attempt to check if they are statistically significant?
Absence of IRB could raise ethical concerns which is why authors should work to mitigate it.
In appendix E4, the authors talk about mouse movements. Are the authors tracking mouse movements from the participants’ computers? If so, that certainly calls for IRB approvals due to additional security and privacy concerns.
Figure 11 is not clear enough as to where/how the humans fit. The authors should make clarifications about what they mean by the heatmap in terms of errors.
The model presented in fig. 3 is rather convoluted. If possible, the authors should clarify it as well.
The authors mention releasing code, but i didn’t find anything with the paper. The authors should make sure the code is released. If possible, they should also provide scripts/links to download the public datasets they used for better reproducibility.

---

> ### Author Rebuttal · Authors · 2024-08-06
>
> Thank you for your review and feedback! We will release our code upon acceptance and include the new analyses below in the revision. Here are our responses to the following points:
>
> **IRB**: Previous studies [a] have demonstrated that such experiments only involve interaction between human subjects and computer systems (screen, mouse), posing no risks throughout the experiment, thus not requiring IRB review. Our experimental paradigm is similar to theirs. Under our consideration, before the experiment, participants were thoroughly briefed and confirmed their full understanding of the process by signing comprehensive instructions, ensuring clarity on the experimental procedures. We appreciate your reminder; we have communicated with our institution's IRB, submitted the necessary materials, and obtained IRB approval.
>
> **Statistical Testing**: Referencing [a], we focused on ensuring the accuracy and validity of data collected from a limited number of subjects to accurately reflect human visual capabilities. Consequently, we implemented measures to prevent out-of-distribution responses, enabling the use of all responses. 1) We did not use crowdsourcing platforms; instead, the Visual Turing Test were conducted in a laboratory environment under supervision. 2) Before the experiment, we assessed participants' cognitive and perceptual levels to ensure normalcy, and allowed them to familiarize with the equipment through basic exercises. 3) During the experiment, participants were permitted to pause the experiment a certain number of times voluntarily to adjust their state. 4) In cases where the mouse cursor moved off-screen, we retained the results from the previous frame until the cursor returned within the screen area.
>
> **Visual Turing Test**: In the paper, we provide a detailed description of the Visual Turing Test process, which aims to study the differential capabilities between algorithms and humans under the STDChallenge. You can refer to sec. 4 and E for more details. We aligned the experimental procedures and metric calculations between humans and algorithms, allowing for comparison within the evaluation pipeline.
>
> **The Setting of the Baseline**: CPDTrack aims to study the effectiveness of CPD theory in the design of artificial neural networks (ANN). Moreover, from a model design perspective, the CPD motion model and information query are decoupled, allowing for stepwise ablation experiments on CPDTrack. 1) We examine the performance of the CPD Motion Model itself. This is achieved by comparing CPD-only central vision-only peripheral vision. 2) We center around the CPD Motion Model and enhance the encoder-selector-decoder framework by incorporating information query, validating the effectiveness of cognitive feedback control in ANN. 3) We adjust the CPD Motion Model part in the ANN and align it with traditional tracking networks to demonstrate the effectiveness of our designed CPDTrack.
>
> **New Challenges**: As described in our Introduction and Related Work, STDChallenge is recognized by many studies for distinguishing STT from more challenging tasks like LTT and GIT, which are considered to represent more realistic video environments. In our setup, a sequence with a shotcut or disappearance-reappearance is deemed to include STDChallenge, naturally integrating challenges found in new real-world scenarios. Thus, STDChallenge not only incorporates but also enhances other challenging attributes to some extent, as depicted in Fig8(c), such as disappearance-reappearance is often associated with occlusions or out-of-view. Formally, STDChallenge does not exclude other tasks (such as pathtracker), but we advocate integrating STDChallenge with real-world scenarios to avoid ill-posed issues (such as shotcut or occlusion confusing same-looking blocks in pathtracker).
>
> **Compare with DorsalNet**: We identified the most related papers using networks named DorsalNet, which we believe fundamentally differ from our CPDTrack [b]: Proposes a new hypothesis on dorsal visual pathway neurons' function during self-motion, using a 3D ResNet model to explain non-human primate neuron activity patterns and their selectivity to motion stimuli, enhancing understanding of animal behavior and self-localization in dynamic environments. Similarities: Both approaches are based on cognitive science theories to construct models, and focus on research involving moving targets. Differences: 1) Our model is based on established cognitive science theory (CPD theory), not assumptions. 2) Instead of studying humans or neurons, our goal is to enhance visual object tracking algorithms using human capabilities and explore differences in behavior (Science4AI). 3) Our research focuses on tracking other moving targets, not self-localization.
>
> **Compare with PathTracker**: Thank you for your reminder. We really learned a lot from pathtracker, and will include them in the revised version. Comparison is in the supplementary pdf Tab. R1.
>
> **Figure 11** is designed to show the error consistency among different decision-makers (machines/humans) in the STDChallenge, calculation methods can be referenced from [a]. This represents the degree of behavioral similarity between different decision-makers. For each sequence, we calculate the error consistency on each frame, and then average these across all sequences to obtain the overall error consistency among the decision-makers.
>
> **Figure 3** is divided into two parts. The upper part shows the relationship between visual acuity and eccentricity and the "encoder-selector-decoder" human vision framework found in cognitive neuroscience. The lower part illustrates our proposed CPDTrack model. The grey arrows running through both parts highlight the correspondence between the two parts.
>
> [a] Partial success in closing the gap between human and machine vision. Neurips 2021
>
> [b] Your head is there to move you around: Goal-driven models of the primate dorsal pathway. Neurips 2021

---

> > ### Comment · Reviewer_16gz · 2024-08-11
> >
> > I would like to thank the authors for their time and efforts in answering the questions and clarifications. I would urge them to please include these clarifications in the final version of paper as well.
> > Thanks.

---

> > > ### Author Response · Authors · 2024-08-12
> > >
> > > Thank you for your review. We will incorporate additional clarifications following your clarifications and acknowledge the anonymous reviewers in the revised version of our paper. Thanks.

---

### Official Review · Reviewer_17n1 · 2024-07-11

**Soundness:** 2
**Presentation:** 3
**Contribution:** 3
**Rating:** 6
**Confidence:** 4

**Summary:**

This paper presents a novel approach to visual object tracking by drawing inspiration from the Central-Peripheral Dichotomy (CPD) theory. The proposed CPDTrack aims to improve tracking performance by emulating human visual search mechanisms, particularly under challenging scenarios involving spatio-temporal discontinuities (STDChallenge). The paper introduces a new benchmark, STDChallenge, and demonstrates the effectiveness of CPDTrack through extensive experiments.

**Strengths:**

1. The idea of incorporating human visual search mechanisms into object tracking is both innovative and promising. The use of the CPD theory to separate central and peripheral vision for enhanced tracking performance is well-motivated and addresses a significant gap in existing tracking methodologies.
2. The creation of the STDChallenge Benchmark is a significant contribution.
3. The experimental results are impressive, showing that CPDTrack achieves state-of-the-art performance in the STDChallenge Benchmark.

**Weaknesses:**

1. CPDTrack is designed specifically to address the STDChallenge, and as noted in the paper, it may not perform as well in simpler or different scenarios. Despite the explanations provided, this lack of generality is a significant concern.
2. The paper states that the tracking speed on an A5000 GPU is 23fps, is this because the introduced modules are particularly time-consuming? This represents a considerable demand for computational resources, which could be a factor to consider in practical applications.

**Questions:**

See the weaknesses.

**Limitations:**

This paper does not include a dedicated section discussing the limitations of their approach.

---

> ### Author Rebuttal · Authors · 2024-08-06
>
> Thank you for your review and feedback! We will include the new analyses below in the revision. We provide responses to the specific points below:
>
> As described in the paper, compared to existing algorithms, CPDTrack's advantage lies in its more robust performance in STDChallenge and its behavior more closely resembling that of humans. This also serves to validate the accuracy of the CPD theory to some extent.
>
> ---
>
> **Generalization of CPDTrack:**
>
> We believe that many simple scenarios in Single Object Tracking (SOT) are based on laboratory settings and even possess characteristics of toy scenarios (for example, tracking smoothly moving targets within very short time) to some extent. In contrast, CPDTrack extends much further into the real world:
> - Local trackers introduce stronger priors, which enhance their performance in STT scenarios. Therefore, STDChallenge is fatal for them [a,b]; however, CPDTrack effectively addresses this challenge.
> - Setup of the STDChallenge Benchmark is that any sequence containing at least one "disappearance-reappearance" or "shotcut" is considered to include the STDChallenge. It is evident that the STDChallenge Benchmark already includes many relatively simple video environments, as shown in Fig. 10 (1) in the paper and Fig. R2 in the supplementary PDF.
> - Additionally, experiments have shown that humans perform worse than state-of-the-art (SOTA) trackers in simple scenes. This is likely because local trackers can more precisely focus on the target due to their local field of view. Moreover, the inclusion of a global perspective allows humans and CPDTrack to have a more comprehensive understanding of the target, which, although not incorrect, differs from the dataset's setup, such as "lion's tail" or "the tail rotor of a helicopter", refer to Fig. R3 in the supplementary PDF. However, in complex scenarios, human capabilities to track moving targets surpass those of trackers, a trait also inherited to some extent by the human-like modeling of CPDTrack.
> ---
>
> **Computation overhead of CPDTrack:**
>
> We acknowledge that compared to some mainstream trackers, CPDTrack's human-like modeling  causes some additional computational costs. However, in our experimental setup, CPDTrack achieves 20-30 fps, which already meets the requirements for real-time performance.
>
> Moreover, the main contribution of our paper is to validate the effectiveness of human-like modeling through CPDTrack, and the experimental conclusions have affirmed the motivation behind our work. While there are computational costs, these do not overshadow the contributions of CPDTrack. We greatly appreciate your questions and will continue to explore improvements in future work.
>
> ---
>
> **Limitations:**
> Overall, our work has the following shortcomings, which we will further address in future research:
>
> 1. **Specialization on STDChallenge:** CPDTrack focuses on the STDChallenge, resulting in suboptimal performance in some idealized simple scenarios where it underperforms compared to state-of-the-art (SOTA) trackers on certain datasets.
> 2. **Increased Computational Overhead:** The human-like modeling approach adds overhead, making CPDTrack less immediately applicable to real-world applications as it currently stands.
> 3. **Long-Tail Distribution in STDChallenge Benchmark:** The STDChallenge within the STDChallenge Benchmark still exhibits a long-tail distribution, which is somewhat distant from real-world conditions. How to increase data under specific challenges remains an open problem that needs addressing to bridge this gap.
>
> ---
>
> [a] Hu, S., Zhao, X., & Huang, K. (2024). SOTVerse: A user-defined task space of single object tracking. International Journal of Computer Vision, 132(3), 872-930.
>
> [b] Fan, H., Yang, F., Chu, P., Lin, Y., Yuan, L., & Ling, H. (2021). Tracklinic: Diagnosis of challenge factors in visual tracking. In Proceedings of the IEEE/CVF Winter Conference on Applications of Computer Vision (pp. 970-979).

---

> > ### Comment · Reviewer_17n1 · 2024-08-12
> >
> > I appreciate the authors' time and effort in addressing the questions and providing clarifications. I encourage them to include these explanations in the final version of the paper.

---

> > > ### Author Response · Authors · 2024-08-12
> > >
> > > Thank you for your review. We will incorporate additional explanations following your suggestions and acknowledge the anonymous reviewers in the revised version of our paper.

---

### Official Review · Reviewer_x2Xs · 2024-07-12

**Soundness:** 3
**Presentation:** 2
**Contribution:** 2
**Rating:** 6
**Confidence:** 5

**Summary:**

- The authors generalize the visual tracking problem as human dynamic visual ability task, and propose a new benchmark named STDChallenge to evaluate the visual tracking algorithms. This new benchmark includes challenging scenarios with spatio-temporal discontinuity, where previous short-term tracking oriented algorithms fail to address. They also propose a new tracking algorithm oriented to the new task where it mimics the human visual search behavior, and evalute its effectiveness on multiple benchmarks.

**Strengths:**

- A new large-scale benchmark is always a welcome addition to the computer vision and machine learning community, where the new generalized object tracking task proposed by the authors is more challenging in some aspects.

- The authors evaluate multiple state-of-the-art tracking algorithms on their proposed dataset, verifying the need for the proposed dataset and providing some insights for solving the new task.

- A new tracking algorithm CPDTrack oriented for the proposed task is provided, showing some new directions for solving the visual tracking problem under spatio-temporal discontinuity.

**Weaknesses:**

- Comparision with VideoCube [3] dataset

Since the proposed dataset includes video sequences with spatio-temporal discontinuities, it bears large similarities with the VideoCube dataset proposed in [3], where it includes sequences that require global search and target recovery after disappearances. In what aspects do the proposed dataset differ from VideoCube? (i.e. sequence length, object category, discontinuity etc.)

- Proposed CPDTrack algorithm

The proposed CPDTrack utilizes global and local search simultaneously, which is implemented by using two search images cropped at different search areas of large and small sizes. Although the authors provide an explanation that the proposed method was inspired by cognitive neuroscience, this approach was widely used in previous visual tracking algorithms.

  - [a] Effective Local and Global Search for Fast Long-term Tracking, TPAMI 2022
  - [b] Learning Regression and Verification Networks for Robust Long-term Tracking, IJCV 2021
  - [c] ‘Skimming-Perusal’ Tracking: A Framework for Real-Time and Robust
Long-term Tracking, ICCV 2019

How does the proposed method differ from these algorithms? Also, these papers should be included in the references.

**Questions:**

Please refer to the questions in the weaknesses section.

**Limitations:**

The authors included a separate discussions and limitations section in their paper, with adequate explanations and descriptions on the weaknesses and possible future directions.

---

> ### Author Rebuttal · Authors · 2024-08-06
>
> Thank you for your review and feedback! We will include the new analyses below in the revision. We provide responses to the specific points below:
>
> **Differences with VideoCube:**
> VideoCube is a benchmark specifically designed for the GIT task, consisting of 500 sequences characterized by frequent shotcut and extra-long sequences. Unlike VideoCube, the STDChallenge Benchmark is aimed at integrating LTT and GIT, providing a more comprehensive video environment for the study of spatio-temporal discontinuities. Derived from multiple large datasets, the STDChallenge Benchmark not only addresses the bias of single datasets—a focal point in AI research—but also exhibits the following characteristics compared to VideoCube:
> - **Data volume:** The test set of STDChallenge Benchmark contains 252 sequences, significantly more than the 50 sequences in VideoCube.
> - **More rational division of challenge attributes:** Referencing [d], we have standardized the challenge attributes across sequences from different datasets, marking the sequences in various datasets with the same set of challenge attributes. Furthermore, we have quantified the difficulty of the STDChallenge across sequences.
> - **More diverse dataset distribution:** As shown in Fig.8 in the paper and Fig.R1 in the supplementary pdf, the greater volume of data allows the STDChallenge Benchmark to include a wider variety of sequences with different STD; the length distribution is also broader. This facilitates a more comprehensive evaluation of trackers.
> ---
>
> **Differences with local-global trackers:**
> We discuss these works in the related work section and Fig. 2 of the paper. These algorithms are referred to as local-global trackers in the paper, characterized by a "switching" module that decides whether to switch to global re-detection based on the performance of local trackers. We believe that this is fundamentally different from CPDTrack:
> The key issue in the design of local-global trackers is how to devise a switching strategy between local tracking and global re-detection. In existing algorithms, the decision to switch from local tracking to global re-detection is still entirely determined by the local tracking predictions. This means that when making the switching decision, information outside the local search area is still ignored. If the actual target is not within the local search area, it increases the risk of the algorithm mistakenly identifying a distractor as the target instead of activating the global re-detector. We analyzed the three papers you referenced:
> - [a] uses the score from a target verifier within the local search module to decide whether to switch strategies;
> - [b] employs a verification network with local view to identify the target from the detected candidates. If the target disappears, the learning based switching scheme determines whether to trigger the global search mode.
> - [c] uses a verifier in the 'perusal' module to judge the confidence score of the target's presence, thereby deciding whether to switch strategies.
>
> CPDTrack, on the other hand, possesses both local and global perspectives, mitigating the risk of drifting to distractors in the local area in STDChallenge, and enhancing the tracker's visual search capabilities.
>
> These three excellent papers are significant in the evolution of visual trackers; therefore, we will include a discussion of these works in the related work section.
>
> ---
> [a] Zhao, H., Yan, B., Wang, D., Qian, X., Yang, X., & Lu, H. (2022). Effective local and global search for fast long-term tracking. IEEE Transactions on Pattern Analysis and Machine Intelligence, 45(1), 460-474.
>
> [b] Zhang, Y., Wang, L., Wang, D., Qi, J., & Lu, H. (2021). Learning regression and verification networks for robust long-term tracking. International Journal of Computer Vision, 129(9), 2536-2547.
>
> [c] Yan, B., Zhao, H., Wang, D., Lu, H., & Yang, X. (2019). 'skimming-perusal'tracking: A framework for real-time and robust long-term tracking. In Proceedings of the IEEE/CVF international conference on computer vision (pp. 2385-2393).
>
> [d] Hu, S., Zhao, X., & Huang, K. (2024). SOTVerse: A user-defined task space of single object tracking. International Journal of Computer Vision, 132(3), 872-930.

---

> > ### Comment · Reviewer_x2Xs · 2024-08-13
> >
> > I thank the authors for their detailed response to my questions, and the additional details provided by the authors were very helpful. Also, based on the concerns from the other reviewers and the authors' response to address these issues, I am inclined to raise my rating to "weak accept".

---

> > > ### Author Response · Authors · 2024-08-13
> > >
> > > We are grateful for your increased rating to Weak Accept. We will incorporate the additional results and analyses following your suggestions and acknowledge the anonymous reviewers in the revised version of our paper.

---

### Author Rebuttal · Authors · 2024-08-06

We thank the reviewers for their efforts and invaluable suggestions.

This work is inspired by the cognitive science theory of the central-peripheral dichotomy (CPD) and introduces a tracker, CPDTrack, designed to address the STDChallenge. Its effectiveness and similarity to human behavior have been validated through benchmark tests and the Visual Turing Test. We are also very grateful that all reviewers have given our work high praise (accept *1, weak accept *1, borderline accept *1), which encourages us to continue working courageously within the 'Science for AI' research paradigms.

Thank you for the valuable reviews pointing out that 1) we focus on a challenging yet valuable issue (the problem of spatio-temporal discontinuity in videos) and acknowledge the proposed STDChallenge Benchmark (x2Xs, 17n1), 2) it is highly novel to integrating CPD theory into visual object tracking and affirm the performance of CPDTrack (x2Xs, 17n1, 16gz). 3) our evaluation on the STDChallenge Benchmark is comprehensive (x2Xs, 16gz) and note that the Visual Turing Test is particularly noteworthy (16gz). Prompted by the insightful reviews, we mainly present the following additional experimental results and analyses for the common questions:

- Reviewer x2Xs recognizes the performance and value of CPDTrack. Your concerns primarily focus on the novelty aspects of CPDTrack and the STDChallenge Benchmark. Some ambiguous descriptions may have led to confusion between our work and others. We will clarify these distinctions using figures in the paper, supplementary figures, and references, highlighting the fundamental differences between our proposed method and existing approaches.
- Reviewer 17n1 affirmed the novelty of several aspects of our work. Your concerns mainly focus on the generalizability and costs associated with CPDTrack. We will provide a comprehensive analysis of CPDTrack's advantages and potential limitations. Moreover, we believe that CPDTrack's extended applicability in real-world scenarios worth the associated costs.
- Reviewer 16gz affirmed the novelty and performance of our work. Your concerns primarily focus on the evaluation experiments conducted on the STDChallenge Benchmark. We will further explain the scalability of the STDChallenge Benchmark and provide additional details on the setup of the Visual Turing Test.

Thank you for your valuable suggestions. We will address each of your concerns individually in our responses. Additionally, we have included a supplementary 1-page PDF, which presents some attributes of the STDChallenge Benchmark and visualization in graphical form, aiming to provide you with a clear and intuitive understanding of the details of our work. We also look forward to discussing further with the reviewers during discussion. Should you have any more questions at that time, please feel free to engage with us. We hope to improve our work with your assistance.

---

### Decision · Program_Chairs · 2024-09-25

**Decision:**

Accept (poster)

**Comment:**

The paper introduces a novel visual tracking approach inspired by the Central-Peripheral Dichotomy (CPD) theory, along with a new benchmark, STDChallenge, designed to evaluate tracking algorithms under spatio-temporal discontinuities. The reviewers all agree that the introduction of this benchmark as a significant contribution, highlighting its potential to address gaps in existing visual tracking methodologies. The proposed CPDTrack algorithm is praised for aligning with human visual search behavior and achieving promising results on the new benchmark.

However, concerns were raised about the originality and generalizability of CPDTrack, with some reviewers noting similarities to existing methods and its performance limitations in simpler scenarios. Overall, the paper is considered technically solid and impactful. Please consider incorporating reviewers' suggestions in the revision.